# DiffusionBlend: Learning 3D Image Prior through Position-aware Diffusion Score Blending for 3D Computed Tomography Reconstruction

**Bowen Song** *     **Jason Hu**\*     **Zhaoxu Luo**     **Jeffrey A. Fessler**     **Liyue Shen**

Department of Electrical and Computer Engineering
University of Michigan
Ann Arbor, MI 48109
`{bowenbw, jashu, luozhx, fessler, liyues}@umich.edu`

## Abstract

Diffusion models face significant challenges when employed for large-scale medical image reconstruction in real practice such as 3D Computed Tomography (CT). Due to the demanding memory, time, and data requirements, it is difficult to train a diffusion model directly on the entire volume of high-dimensional data to obtain an efficient 3D diffusion prior. Existing works utilizing diffusion priors on single 2D image-slice with hand-crafted cross-slice regularization would sacrifice the z-axis consistency, which results in severe artifacts along the z-axis. In this work, we propose a novel framework that enables learning the 3D image prior through position-aware 3D-patch diffusion score blending for reconstructing large-scale 3D medical images. To the best of our knowledge, we are the first to utilize a 3D-patch diffusion prior for 3D medical image reconstruction. Extensive experiments on sparse view and limited angle CT reconstruction show that our DiffusionBlend method significantly outperforms previous methods and achieves state-of-the-art performance on real-world CT reconstruction problems with high-dimensional 3D image (i.e., $256 \times 256 \times 500$). Our algorithm also comes with better or comparable computational efficiency than previous state-of-the-art methods. Code is available at: `https://github.com/efzero/DiffusionBlend`.

## 1   Introduction

Diffusion models learn the prior of an underlying data distribution, which enables sampling from the distribution to generate new images [1–3]. By starting with a clean image and gradually adding noise of different scales, diffusion sampler eventually obtains an image that is indistinguishable from pure noise. Let $\boldsymbol{x}_t$ be the image sequence where $t = 0$ represents the clean image and $t = T$ is pure noise. The score function of the image distribution, denoted as $\boldsymbol{s}(\boldsymbol{x}_t) = \nabla \log p(\boldsymbol{x}_t)$, can be learned by a neural network parametrization, which takes $\boldsymbol{x}_t$ as input and then approximates $\nabla \log p(\boldsymbol{x}_t)$. The reverse process then starts with pure noise and uses the learned score function to iteratively remove noise, ending with a clean image sampled from the target distribution $p(\boldsymbol{x})$.

Leveraging the learned score function as a prior, it is efficient to solve the inverse problems based on diffusion priors. Previous works have proposed to use diffusion inverse solvers for deblurring, super-resolution, and medical image reconstruction such as in magnetic resonance imaging (MRI) and computed tomography (CT), and many other applications [4–16].

---

*These authors contributed equally to the work

38th Conference on Neural Information Processing Systems (NeurIPS 2024).

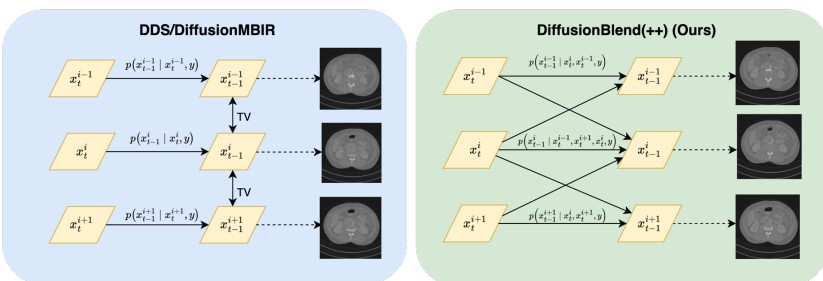

Figure 1: Overview of DiffusionBlend++ compared to previous 3D image reconstruction works. Previous work used a hand-crafted TV term to "regularize" adjacent slices, whereas the proposed approach uses learned diffusion score blending between groups of slices. Here $i$ is the slice index, and $t$ is the reconstruction iteration.

Computed tomography (CT) reconstruction is an important inverse problem that aims at reconstructing the volumetric image $x$ from the measurements $y$, which is acquired from the projections at different view angles [17]. To reduce the radiation dose delivered to the patient, sparse-view CT uses a smaller fraction of X-rays compared to the full-view CT [18]. Additionally, limited-angle CT is useful in cases where patients may have mobility issues and cannot use full-angle CT scans [19]. Although previous works have discussed and proposed diffusion-based methods for solving the 2D CT image reconstruction problem to demonstrate the proof-of-concept [9, 10], there is very limited work focusing on solving inverse problems for 3D images due to the practical difficulty in capturing 3D image prior. Learning efficient 2D image priors using diffusion models is already computationally expensive, which requires large-scale of training data, training time, and GPU memory. For example, previous works [2, 3] require training for several days to weeks on over a million training images in the ImageNet [20] and LSUN [21] datasets to generate high-quality 2D natural images of size $256 \times 256$. Hence, directly learning a 3D diffusion prior on the entire CT volume would be practically infeasible or prohibitively expensive due to the demanding requirements of training data and computational cost. In addition, real clinical CT data is usually limited and scarce and often has a resolution larger than $256 \times 256 \times 400$, which makes directly training the data prior very challenging. The problem of tackling 3D image inverse problems, especially for medical imaging remains a challenging open research question.

A few recent works [13–15] have discussed and proposed to solve 3D image reconstruction problems either through employing some hand-crafted regularization to enforce consistency between 2D slices when reconstructing 3D volumetric images [13, 15], or through training several diffusion models for 2D images on each plane (axial, coronal, and sagittal), and performing reverse sampling with each model alternatively [14]. However, all of these works only learn the distribution of a single 2D slice via the diffusion model, while having not yet explored the dependency between slices that is required to better model the real 3D image prior.

To overcome these limitations, we propose a novel method, called DiffusionBlend, that enables learning the distribution of 3D image patches (a batch of nearby 2D slices), and blends the scores of patches to model the entire 3D volume distribution for image reconstruction. Specifically, we firstly propose to train a diffusion model that models the joint distribution of 3D image patches (nearby 2D slices) in the axial plane conditioning on the slice thickness. Then, we introduce a random blending algorithm that approximates the score function of the entire 3D volume by using the trained 3D-patch score function. Moreover, we can either directly use the trained model to predict the noise of a single 2D slice by taking its corresponding 3D patch as input, or applying a random blending algorithm that firstly randomly partitions the volume into different 3D patches at each time step and then computes the score of each 3D patch during reverse sampling. Through either way, we can output the predicted noise of the entire 3D volume. In this way, our proposed method is able to enforce cross-slice consistency without any hand-crafted regularizer. Our method has the advantage of being fully data-driven and can enforce slice consistency without the TV regularizer as demonstrated in Fig. 1. Through exhaustive experiments of ultra-sparse-view and limited-angle 3D CT reconstruction on different datasets, we validate that our proposed method achieves superior reconstruction results for 3D volumetric imaging, outperforming previous state-of-the-art (SOTA) methods. Furthermore, our method achieves better or comparable inference time than SOTA methods, and requires minimum hyperparameter tuning for different tasks and settings.

In summary, our main contributions are as follows:

- We propose DiffusionBlend(++): a novel method for 3D medical image reconstruction through 3D diffusion priors. To the best of our knowledge, our method is the first diffusion-based method that learns the 3D-patch image prior incorporating the cross-slice dependency, so as to enforce the consistency for the entire 3D volume without any external regularization.

- Specifically, instead of independently training a diffusion model only on separated 2D slices, we propose a novel method that first trains a diffusion model on 3D image patches (a batch of nearby 2D slices) with positional encoding, and at inference time, employs a new approach of random partitioning and diffusion score blending to generate an isotropically smooth 3D volume.

- Extensive experiments validate our proposed method achieves **state-of-the-art** reconstruction results for 3D volumetric imaging for the task of ultra-sparse-view and limited-angle 3D CT reconstruction on different datasets, with improved inference time efficiency and minimal hyperparameter tuning.

## 2 Background and Related Work

**Diffusion models.** Diffusion models consists of a forward process that gradually adds noise to a clean image, and a reverse process that denoises the noisy images [1, 22]. The forward model is given by $x_t = x_{t-1} - \frac{\beta_t \Delta t}{2} x_{t-1} + \sqrt{\beta_t} \Delta t \omega$ where $\omega \in N(0,1)$ and $\beta(t)$ is the noise schedule of the process. The distribution of $\boldsymbol{x}(0)$ is the data distribution and the distribution of $\boldsymbol{x}(T)$ is approximately a standard Gaussian. When we set $\Delta t \to 0$, the forward model becomes $dx_t = -\frac{1}{2}\beta_t x_t dt + \sqrt{\beta_t} d\omega_t$, which is a stochastic differential equation. The solution of this SDE is given by

$$dx_t = \left( -\frac{\beta(t)}{2} - \beta(t) \nabla_{x_t} \log p_t(x_t) \right) dt + \sqrt{\beta(t)} d\overline{\boldsymbol{w}}. \tag{1}$$

Thus, by training a neural network to learn the score function $\nabla_{x_t} \log p_t(x_t)$, one can start with noise and run the reverse SDE to obtain samples from the data distribution.

Although diffusion models have achieved impressive success for image generation, a bottleneck of large-scale computational requirements including demanding training time, data, and memory prevents training a diffusion model directly on high-dimensional high-resolution images. Many recent works have been studying how to improve the efficiency of diffusion models to extend them to large-scale data problem. For example, to reduce the computational burden, latent diffusion models [23] have been proposed, aiming to perform the diffusion process in a much smaller latent space, allowing for faster training and sampling. However, solving inverse problems with latent diffusion models is still a challenging task and may have sub-par computational efficiency [24]. Very recently, various methods have been proposed to perform video generation using diffusion models, generally by leveraging attention mechanisms across the temporal dimension [25–28]. However, these methods only focus on video synthesis. Utilizing these complicated priors for posterior sampling is still a challenge because if these methods were applied to physical 3D volumes, continuity would only be maintained across slices in the XY plane and not the other two planes. Finally, work has been done to perform sampling faster [29–31], which is unrelated to the training process and network architecture. However, although these methods effectively promote the efficiency of training a diffusion model, current works are not yet able to tackle the large-scale 3D image reconstruction problem in real world settings.

**3D CT reconstruction** Computed tomography (CT) is a medical imaging technique that allows a 3D object to be imaged by shooting X-rays through it [17]. The measurements consist of a set of 2D projection views obtained from setting up the source and detector at different angles around the object. By definition, $\boldsymbol{y}$ is the (known) set of projection views, $\mathcal{A}$ is the (in most cases assumed to be) linear forward model of the CT measurement system, and $\boldsymbol{x}$ is the unknown image. The CT reconstruction problem then consists of reconstructing $\boldsymbol{x}$ given $\boldsymbol{y}$. Traditional methods for solving this include regularization-based methods that enforce a previously held belief on $\boldsymbol{x}$ and likelihood based methods [17, 32–34].

Data-driven methods have shown tremendous success in signal and image processing in recent years [35–39]. In particular, for solving inverse problems, when large amounts of training data

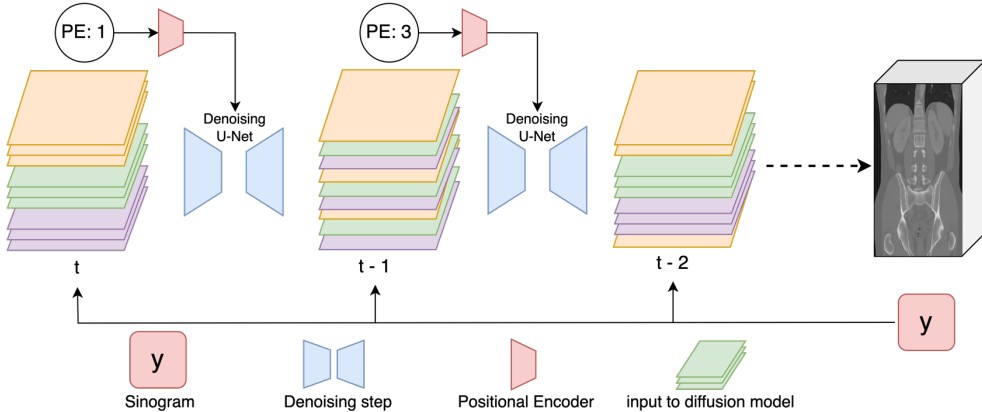

Figure 2: Overview of slice blending process during reconstruction for DiffusionBlend++. At each iteration, we partition the slices of the volume in a different way; slices of the same color are inputted into the network independently. Positional encoding (PE) is also inputted to the network as information about the separation between the slices.

is available, a learned prior can be much stronger than the hand-crafted priors used in traditional methods [40, 41]. For past few years, many deep learning-based method have been proposed for solving the 3D CT reconstruction problem [42–45]. These methods train a convolutional neural network, such as a U-Net [42], that maps the partial-view filtered backprojection (FBP) reconstructed image to the ground truth image, that is, full-view CT reconstruction. However, these methods often generate blurry images and generalizes poorly for out-of-distribution data [46].

**3D CT reconstruction with diffusion models.** Diffusion models serve as a very strong prior as they can generate entire images from pure noise. Most methods that use diffusion models to solve inverse problems formulate the task as a conditional generation problem [47–49] or as a posterior sampling problem [4–6, 9, 50]. In the former case, the network requires the measurement $y$ (or an appropriate resized transformation of $y$) during training time. Thus, at reconstruction time, that trained network can only be used for solving the specific inverse problem with poor generalizability. In contrast, for the posterior sampling framework, the network learns an unconditional image prior for $x$ that can help solve various inverse problem related to $x$ without retraining. Although these diffusion-based methods have shown great performance for solving inverse problems for 2D images in different domains, there are seldom methods that are able to tackle inverse problems for 3D images because of the infeasible computational and data requirements as aforementioned. Specifically, for 3D CT reconstruction, DiffusionMBIR [13] trains a diffusion model on the axial slices of volumes; at reconstruction time, it uses the total variation (TV) regularizer with a posterior sampling approach to encourage consistency between adjacent slices. Similarly, DDS [15] builds on this work by using accelerated methods of sampling and data consistency to greatly reduce the reconstruction time. However, although the TV regularizer has shown some success in maintaining smoothness across slices, it is not a data-driven method and does not properly learn the 3D prior. TPDM [14] addresses this problem by training a separate prior on the coronal slices of volumes with a conditional sampling approach, which serves as a data-driven method of maintaining slice consistency at reconstruction time, but requires that all the volumes have the same cubic shape. In exchange, this method sacrifices the speed gains made by DDS, requiring alternating updates between the two separate priors, and is also twice as computationally expensive at training time. To overcome these limitations, we aim to propose a more flexible and robust approach that can learn the 3D data prior properly for CT reconstruction, maintaining slice consistency while not sacrificing inference time.

## 3   Methods

Instead of modeling the 2D slices of the 3D volume as independent data samples during training time, and then applying regularization between slices at reconstruction time, we propose incorporating information from neighboring slices at training time to enforce consistency between slices. More precisely, our first approach models the data distribution of a 3D volume with $H$ slices in the $z$

dimension as follows:

$$p(\boldsymbol{x}) \approx \prod_{i=1}^{H} p(\boldsymbol{x}[:, :, i] \mid \boldsymbol{x}[:, :, i - j : i - 1], \boldsymbol{x}[:, :, i + 1 : i + j])/Z, \qquad (2)$$

where $j$ is a positive integer indicating the number of neighboring slices above and below the target slice that are being used as conditions to predict the target slice, and $Z$ is a normalizing constant. To deal with boundary conditions where the third index may exceed the bounds of the original volume, we apply repetition padding above and below the main volume.

For training, we simply concatenate each of the conditioned slices with the target slice along the channel dimension to serve as an input to the neural network. Then we apply denoising score matching to predict the noise of the target slice as the loss function of the neural network:

$$\mathbb{E}_{t \sim \mathcal{U}(0,T)} \mathbb{E}_{\boldsymbol{x} \sim p(\boldsymbol{x})} \mathbb{E}_{\epsilon \sim \mathcal{N}(0,I)} \mathbb{E}_{i \in [1,H]} \|\epsilon_\theta(\mathbf{x}_t[:, :, i - j : i + j], \sigma_t) - \epsilon[:, :, i]\|_2^2. \qquad (3)$$

At reconstruction time, the score function of the entire volume decomposes as a sum of score functions of each of the slices:

$$\nabla \log p(\boldsymbol{x}) \approx \sum_{i=1}^{H} \nabla \log p(\boldsymbol{x}[:, :, i] \mid \boldsymbol{x}[:, :, i - j : i - 1], \boldsymbol{x}[:, :, i + 1 : i + j]). \qquad (4)$$

In this way, we have rewritten the score of the 3D volume as sums of the scores of the 2D slices learned by the network. This means that we can now apply any algorithm that uses diffusion models to solve inverse problems to solve the 3D CT reconstruction problem. Furthermore, this method of blending together information from different slices allows us to learn a prior for the entire volume that combines information from different slices. We call this method **DiffusionBlend**.

To learn an even better 3D image prior, instead of learning the conditional distribution of individual target slices, we can learn the **joint distribution** of several neighboring slices at once, which we call a 3D patch. Letting $k$ be the number of slices in each patch, we can partition the volume into 3D patches and approximate the distribution of the volume as

$$p(\boldsymbol{x}) \approx (\prod_{i=1}^{H/k} p(\boldsymbol{x}[:, :, (i - 1)k + 1 : ik]))/Z, \qquad (5)$$

where $Z$ is a normalizing constant. Comparing this with (2), the main difference is instead of conditioning on neighboring slices, we are now incorporating the neighboring slices as a joint distribution. This allows for much faster reconstruction, as $k$ slices are updated simultaneously according to their score function. However, this method faces similar slice consistency issues as in [13], since certain pairs of adjacent slices (namely, pairs whose slice indices are congruent to 0 and 1 modulo $k$) are never updated simultaneously by the network.

To deal with this issue, we propose two additional changes. Firstly, instead of using the same partition (updating the same $k$ slices) at once for each iteration, we can use a different partition so that the previous border slices can be included in another partition. For example, we can randomly sample the end index of the first 3D patch for adjacency slices. Let $m$ be uniformly sampled from $1, 2, ..., k$, we can use the partition

$$\mathcal{S} = \{1, 2, \ldots, H\} = \{1, \ldots, m\} \cup \{m + 1, \ldots, m + k\} \cup \ldots \cup \{H - k + 1, \ldots, H\}, \quad (6)$$

instead of $\mathcal{S} = \{1, 2, \ldots, H\} = \{1, \ldots, k\} \cup \{k+1, \ldots, 2k\} \cup \ldots \cup \{H - k + 1, \ldots, H\}$, where $m$ is the offset index number in the new partition. We can then compute the score on the new partition. More generally, we can choose an arbitrary partition of $\mathcal{S}$ into $H/k$ sets, each containing $k$ elements for each iteration, updating each slice in the small set simultaneously for that iteration.

Secondly, to better capture information between nonadjacent slices, we apply relative positional encoding as an input to the network. More precisely, if a 3D patch has a slice thickness (the distance between two slices) of $p$, then we let $p$ be input of the positional encoding for that 3D patch. The positional encoding block consists of a sinusoidal encoding module and several dense connection modules, which has the same architecture as the timestamp embedding module of the same diffusion model. In this manner, the network is able to learn how to incorporate information from nonadjacent slices and captures more global information about the entire volume. Recall that for 3D patches of adjacent slices, the border between patches may have inconsistencies. To address this, we can **concatenate each border** as a new 3D patch, and then compute the score from it. If there are $k$ slices in an adjacency-slice 3D patch, then the new 3D patch has the relative positional encoding of $k$, and also has a size of $k$. For instance, if the previous partition is (1,2,3),(4,5,6),(7,8,9), the new partition

is (1,4,7),(2,5,8),(3,6,9). Here we are forming a new partition with jumping slices. In practice, since we need a pretrained natural image checkpoint due to scarcity of medical image data, we set $k = 3$ for facilitating fine tuning from natural image checkpoints.

We call the partitioning by 3D patch with adjacent slices as **Adjacency Partition**, and the partitioning by 3D patch with jumping slices as **Cross Partition**. Letting $r = H/k$ be the number of 3D patches, with a random partition, this method is stochastically averaging the different estimations of the $\nabla \log p(\boldsymbol{x})$ by different paritiions. Specifically, the estimation of score by a single partition $\mathcal{S}_1 \cup \ldots \cup \mathcal{S}_r$ is given by $\sum_{i=1}^{r} \nabla \log p(\boldsymbol{x}[:, :, \mathcal{S}_i])$. Ideally, we want to compute

$$|S|^{-1} \sum_{\mathcal{S}=\mathcal{S}_1 \cup \ldots \cup \mathcal{S}_r} \sum_{i=1}^{r} \nabla \log p(\boldsymbol{x}[:, :, \mathcal{S}_i]). \tag{7}$$

Similar to [4, 13, 51], we can share the summation in (7) across different diffusion steps since the difference between two adjacent iterations $\boldsymbol{x}_i$ and $\boldsymbol{x}_{i+1}$ is minimal.

In summary, we have shown how the score function of the entire volume can be written in terms of scores of the slices of the volume. Hence, similar to DiffusionBlend, this method can be coupled with any inverse problem solving algorithm. The scores of the slices can be approximated using a neural network. Training this network consists of randomly selecting $k$ slices from a volume and concatenating them along the channel dimension to get the input to the network (along with the positional encoding of the slices), and then using denoising score matching as in (3) as the loss function; Section A.1 provides a theoretical justification for this procedure.

**Sampling and reconstruction.** With Eq. 7, each reconstruction step would require computing the score functions corresponding to each of the partitions of $\mathcal{S}$, and then summing them to get the score function $\boldsymbol{s}(\boldsymbol{x})$. We propose the variable sharing technique for this method, and only need to compute the score of one partition per time step. Hence, each iteration, we instead randomly choose one of the partitions of $\mathcal{S}$ and update the volume of intermediate samples by the score function. Finally, we use repetition padding if $H$ is not a multiple of $k$. This method incorporates a similar slice blending strategy as DiffusionBlend, but allows for significant acceleration at reconstruction time as $k$ slices are updated at once. Furthermore, it allows the network to learn joint information between slices that are farther apart without requiring the increase in computational cost associated with increasing $k$. We call this method **DiffusionBlend++**. The pseudocode of the algorithm can be found in Alg. 1.

In practice, we choose not to select from all possible partitions, but instead select from those where the indices in each $\mathcal{S}_i$ are not too far apart, as the joint information between slices that are very far apart is hard to capture. Table 12 summarizes the different 3D image prior models. The appendix provides more details about the partition selection scheme.

**Krylov subspace methods.** Following the work of [15], we apply Krylov subspace methods to enforce data consistency with the measurement. At each timestep $t$, by using Tweedie's formula [52], we compute $\hat{\boldsymbol{x}}_t = \mathbb{E}[\boldsymbol{x}_0 | \boldsymbol{x}_t]$, and then apply the conjugate gradient method

$$\hat{\boldsymbol{x}}'_t = \mathrm{CG}(\boldsymbol{A}^*\boldsymbol{A}, \boldsymbol{A}^*\boldsymbol{y}, \hat{\boldsymbol{x}}_t, M), \tag{8}$$

where in practice, the CG operator involves running $M$ CG steps for the normal equation $\boldsymbol{A}^*\boldsymbol{y} = \boldsymbol{A}^*\boldsymbol{A}\boldsymbol{x}$. We combine this method with the DDIM sampling algorithm [29] to decrease reconstruction time. To summarize, we provide the algorithm for DiffusionBlend++ below. The Appendix provides the training algorithms for our proposed method as well as the reconstruction algorithm for DiffusionBlend.

## 4 Experiments

**Experimental setup.** We used the public CT dataset from the AAPM 2016 CT challenge [53] that consists of 10 volumes. We rescaled the images in the XY-plane to have size $256 \times 256$ without altering the data in the Z-direction and used 9 of the volumes for training data and the tenth volume as test data. The training data consisted of approximately 5000 2D slices and the test volume had 500 slices. We also performed experiments on the LIDC-IDRI dataset [54]. For this dataset, we first applied data preprocessing by setting the entire background of the volumes to zero. We rescaled the images in the XY-plane to have size $256 \times 256$, and, to compare with the TPDM method, only took

---
**Algorithm 1** DiffusionBlend++
---
**Require:** Forward model $\boldsymbol{A}$, sinogram $\boldsymbol{y}$, hyperparameter $k$, CG iteration numbers $M$
    Initialize $\boldsymbol{x}_T \sim \mathcal{N}(0, \sigma_T^2 \boldsymbol{I})$
    **for** $t = T : 1$ **do**
        Randomly select a partition $\mathcal{S} = \mathcal{S}_1 \cup \ldots \cup \mathcal{S}_r$ (if $t \mod k = 0$, then use cross partition, otherwise use random adjacency partitions)
        Compute the relative positional encoding $PE_t$
        For each $i$ compute $\boldsymbol{\epsilon}_\theta(\boldsymbol{x}_t[:, :, \mathcal{S}_i], PE_t)$
        Compute $\boldsymbol{s} = \nabla \log p(\boldsymbol{x}_t)$ using (7)
        Compute $\hat{\boldsymbol{x}}_t = \mathbb{E}[\boldsymbol{x}_0 | \boldsymbol{x}_t]$ using Tweedie's formula
        Set $\hat{\boldsymbol{x}}_t' = \text{CG}(\boldsymbol{A}^*\boldsymbol{A}, \boldsymbol{A}^*\boldsymbol{y}, \hat{\boldsymbol{x}}_t, M)$
        Sample $\boldsymbol{x}_{t-1}$ using $\hat{\boldsymbol{x}}_t'$ and $\boldsymbol{s}$ via DDIM sampling
    **end for**
Return $\boldsymbol{x}$.
---

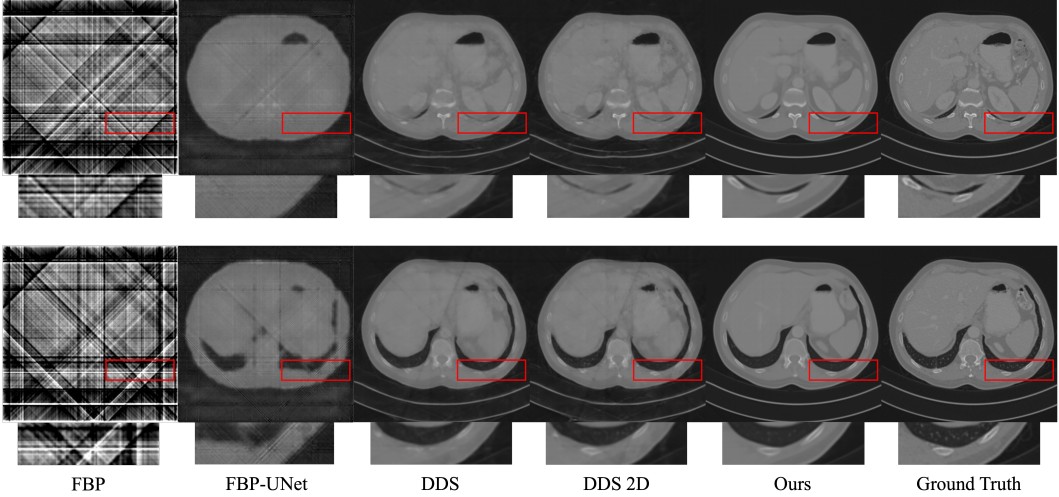

| FBP | FBP-UNet | DDS | DDS 2D | Ours | Ground Truth |

Figure 3: Results of CT reconstruction with 4 views on AAPM dataset, axial view.

the volumes with at least 256 slices in the Z-direction, truncating the Z-direction to have exactly 256 slices. This resulted in 357 volumes which we used for training and one volume used for testing.

We performed experiments for sparse view CT (SVCT) and limited angle CT (LACT). The detector size was set to 512 pixels for all cases. For SVCT, we ran experiments on 4, 6, and 8 views. We also ran additional experiments on 20, 40, 60, 80, and 100 views and report the quantitative results in the Appendix. For LACT, we used the full set of views but only spaced around a 90 degree angle. In all cases, implementations of the forward and back projectors can be found in [13].

For a fair comparison between DiffusionBlend and DiffusionBlend++, we selected $j = 1$ for DiffusionBlend and each $\mathcal{S}_i$ to contain 3 elements for DiffusionBlend++. In this manner, both methods involve learning a prior that involves products of joint distributions on 3 slices. To train the score function for DiffusionBlend, we started from scratch using the LIDC dataset. Since this dataset consisted of over 90000 slices, the network was able to properly learn this prior. We then fine tuned this network on the much smaller AAPM dataset. For DiffusionBlend++, the input and output images both had 3 channels from stacking the slices, so we fine-tuned the existing checkpoint from [22]. All networks were trained on PyTorch using the Adam optimizer with A40 GPUs. For reconstruction, we used 200 neural function evaluations (NFEs) for all the results. The appendix provides the full experiment hyperparameters. We observe that DiffusionBlend++ can reconstruct very high quality images that are free of artifacts as demonstrated in Fig.4 and Fig.3.

**Comparison methods.** We compared our proposed method with baseline methods for CT reconstruction and state of the art 3D diffusion model methods. We used the filtered back projection implementation found in [13]. For the other baseline, we used FBP-UNet [42] which is a supervised method that involves training a network for each specific task mapping the FBP reconstruction to the

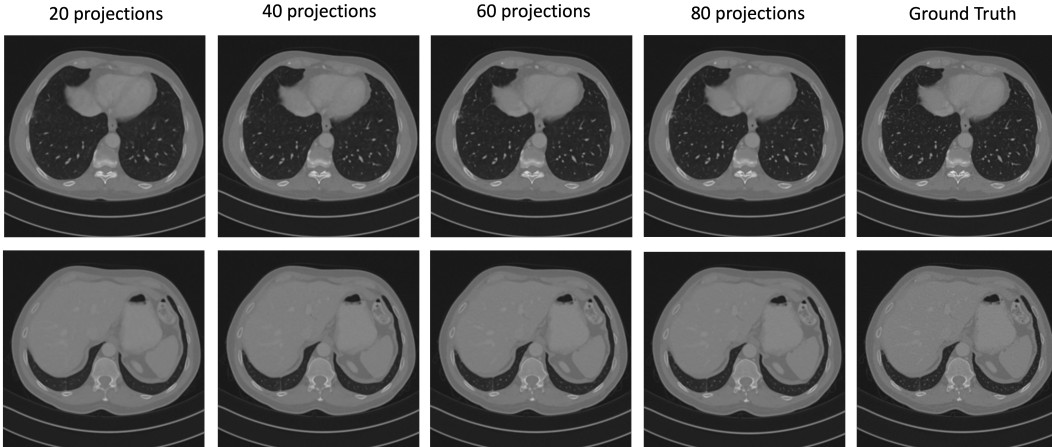

Figure 4: Results of DiffusionBlend++ reconstruction with multiple views on AAPM dataset, axial view.

| Method | Sparse-View CT Reconstruction on AAPM | | | | | | Sparse-View CT Reconstruction on LIDC | | | | | |
| | 8 views | | 6 views | | 4 views | | 8 views | | 6 Views | | 4 Views | |
| | PSNR↑ | SSIM↑ | PSNR↑ | SSIM↑ | PSNR↑ | SSIM↑ | PSNR↑ | SSIM↑ | PSNR↑ | SSIM↑ | PSNR↑ | SSIM↑ |
|---|---|---|---|---|---|---|---|---|---|---|---|---|
| FBP | 14.66 | 0.359 | 13.65 | 0.293 | 11.94 | 0.222 | 14.79 | 0.217 | 14.11 | 0.191 | 13.18 | 0.169 |
| FBP-UNet | 26.00 | 0.849 | 24.15 | 0.782 | 23.37 | 0.761 | 28.58 | 0.848 | 26.48 | 0.781 | 25.19 | 0.731 |
| DiffusionMBIR | 26.30 | 0.863 | 24.99 | 0.827 | 23.66 | 0.789 | 32.67 | 0.922 | 31.18 | 0.901 | 29.02 | 0.863 |
| TPDM | - | - | - | - | - | - | 27.51 | 0.816 | 25.60 | 0.776 | 21.99 | 0.695 |
| DDS 2D | 32.89 | 0.946 | 31.40 | 0.934 | 28.77 | 0.906 | 30.82 | 0.897 | 29.38 | 0.867 | 27.54 | 0.826 |
| DDS | 33.19 | 0.945 | 31.94 | 0.942 | 29.22 | 0.916 | 31.65 | 0.915 | 30.12 | 0.888 | 27.20 | 0.808 |
| DiffusionBlend (Ours) | 34.29 | 0.955 | 33.26 | 0.949 | 31.84 | 0.944 | 33.34 | 0.933 | 30.94 | 0.905 | 27.96 | 0.849 |
| DiffusionBlend++ (Ours) | **35.69** | **0.966** | **34.68** | **0.960** | **32.93** | **0.952** | **34.46** | **0.947** | **33.03** | **0.932** | **30.98** | **0.912** |

Table 1: Comprehensive comparison of quantitative results on Sparse-View CT Reconstruction on Axial View for AAPM and LIDC datasets. Best results are in bold.

clean image. Since this is a 2D method, we learned a mapping between 2D slices and then stacked the 2D slices to get the final 3D volume. We also compared with classical CT reconstruction techniques such as SBTV, SIRT, and CGLS [55] to benchmark our algorithm against traditional methods. Results for these methods are reported in the Appendix. For DiffusionMBIR [56], we fine-tuned the score function checkpoints on our data and used the same hyperparameters as the original work. We did the same for TPDM [14]; however, we ran TPDM only on the LIDC dataset because TPDM requires cubic volumes. Finally, we ran two variants of DDS [15]: one in which all the hyperparameters were left unchanged (DDS), and another in which no TV regularizer between slices was enforced (DDS 2D). Both of these methods were run with 200 NFEs. The appendix provides the experiment parameters.

**Sparse-view CT.** The results for different numbers of views and across different slices are shown in Tables 1, 11, and 3. DiffusionBlend++ exhibits much better performance over all the previous baseline methods (usually by a few dB) and outperforms DiffusionBlend. The second best method for each experiment is underlined and was, in most cases, DiffusionBlend. The exceptions are when the second best method is DiffusionMBIR, but this method was run with 2000 NFEs and took about 20 hours to run compared to 1-2 hours for both of our methods. The two DDS methods required similar runtime as our methods but in all cases exhibited inferior reconstruction results. Furthermore, DDS 2D generally performed worse than DDS. Thus, DDS failed to properly learn a 3D volume prior and still relied on the TV regularizer. Additionally, although TPDM should learn a 3D prior, the results were very poor compared to the other baselines. Our proposed method learned a fully 3D prior and achieved the best results in the sagittal and coronal views.

**Limited-angle CT.** Table 4 shows all results for limited angle CT reconstruction for both the AAPM and LIDC datasets. Our DiffusionBlend++ method obtains superior performance over all the baseline methods and DiffusionBlend obtains the second best results. Similar to the SVCT experiments, DiffusionMBIR performed the best out of the baseline methods, but took approximately 40 hours to

| Method | Sparse-View CT Reconstruction on AAPM | | | | | | Sparse-View CT Reconstruction on LIDC | | | | | |
|---|---|---|---|---|---|---|---|---|---|---|---|---|
| | 8 views | | 6 views | | 4 views | | 8 Views | | 6 Views | | 4 Views | |
| | PSNR↑ | SSIM↑ | PSNR↑ | SSIM↑ | PSNR↑ | SSIM↑ | PSNR↑ | SSIM↑ | PSNR↑ | SSIM↑ | PSNR↑ | SSIM↑ |
| FBP | 12.30 | 0.345 | 10.14 | 0.277 | 6.78 | 0.204 | 14.88 | 0.234 | 14.30 | 0.207 | 13.43 | 0.187 |
| FBP-UNet | 26.13 | 0.860 | 24.14 | 0.798 | 23.47 | 0.779 | 28.56 | 0.848 | 26.52 | 0.783 | 25.29 | 0.732 |
| DiffusionMBIR | 26.64 | 0.869 | 25.08 | 0.834 | 23.71 | 0.789 | 32.79 | 0.922 | 31.30 | 0.900 | 28.98 | 0.862 |
| TPDM | - | - | - | - | - | - | 27.66 | 0.819 | 25.57 | 0.784 | 21.87 | 0.708 |
| DDS 2D | 33.22 | 0.949 | 31.69 | 0.937 | 29.39 | 0.909 | 30.98 | 0.894 | 29.40 | 0.862 | 27.54 | 0.819 |
| DDS | 33.43 | 0.945 | 32.18 | 0.947 | 29.86 | 0.924 | 31.80 | 0.915 | 30.13 | 0.889 | 27.26 | 0.818 |
| DiffusionBlend (Ours) | 35.09 | 0.958 | 33.97 | 0.952 | 32.38 | 0.943 | 33.73 | 0.934 | 31.16 | 0.907 | 27.93 | 0.855 |
| DiffusionBlend++ (Ours) | **36.48** | **0.968** | **35.38** | **0.963** | **33.22** | **0.954** | **34.86** | **0.946** | **33.20** | **0.932** | **30.97** | **0.913** |

Table 2: Comprehensive comparison of quantitative results on Sparse-View CT Reconstruction on Sagittal View for AAPM and LIDC datasets. Best results are in bold.

| Method | Sparse-View CT Reconstruction on AAPM | | | | | | Sparse-View CT Reconstruction on LIDC | | | | | |
|---|---|---|---|---|---|---|---|---|---|---|---|---|
| | 8 views | | 6 views | | 4 views | | 8 Views | | 6 Views | | 4 Views | |
| | PSNR↑ | SSIM↑ | PSNR↑ | SSIM↑ | PSNR↑ | SSIM↑ | PSNR↑ | SSIM↑ | PSNR↑ | SSIM↑ | PSNR↑ | SSIM↑ |
| FBP | 14.64 | 0.325 | 13.18 | 0.268 | 11.16 | 0.236 | 14.78 | 0.206 | 14.10 | 0.181 | 13.10 | 0.165 |
| FBP-UNet | 27.34 | 0.878 | 25.12 | 0.827 | 24.10 | 0.810 | 28.87 | 0.858 | 26.59 | 0.793 | 25.37 | 0.744 |
| DiffusionMBIR | 29.86 | 0.908 | 28.12 | 0.875 | 25.68 | 0.843 | 33.29 | 0.922 | 31.69 | 0.903 | 29.21 | 0.868 |
| TPDM | - | - | - | - | - | - | 28.12 | 0.833 | 25.78 | 0.804 | 22.29 | 0.735 |
| DDS 2D | 33.64 | 0.950 | 32.33 | 0.939 | 30.25 | 0.916 | 31.60 | 0.898 | 29.99 | 0.871 | 28.03 | 0.830 |
| DDS | 33.97 | 0.934 | 32.95 | 0.930 | 30.89 | 0.932 | 32.51 | 0.920 | 30.83 | 0.898 | 27.61 | 0.828 |
| DiffusionBlend (Ours) | 36.45 | 0.958 | 35.23 | 0.952 | 33.98 | 0.944 | 34.47 | 0.934 | 31.48 | 0.908 | 28.24 | 0.859 |
| DiffusionBlend++ (Ours) | **37.87** | **0.968** | **36.66** | **0.963** | **34.27** | **0.955** | **35.66** | **0.947** | **33.97** | **0.935** | **31.38** | **0.913** |

Table 3: Comprehensive comparison of quantitative results on Sparse-View CT Reconstruction on Coronal View for AAPM and LIDC datasets. Best results are in bold.

| Method | AAPM Dataset | | | | | | LIDC Dataset | | | | | |
|---|---|---|---|---|---|---|---|---|---|---|---|---|
| | Axial | | Sagittal | | Coronal | | Axial | | Sagittal | | Coronal | |
| | PSNR↑ | SSIM↑ | PSNR↑ | SSIM↑ | PSNR↑ | SSIM↑ | PSNR↑ | SSIM↑ | PSNR↑ | SSIM↑ | PSNR↑ | SSIM↑ |
| FBP | 16.36 | 0.643 | 16.36 | 0.524 | 15.62 | 0.531 | 18.79 | 0.672 | 19.84 | 0.675 | 20.01 | 0.676 |
| FBP-UNet | 27.38 | 0.910 | 27.81 | 0.918 | 28.44 | 0.930 | 29.42 | 0.885 | 29.50 | 0.884 | 29.54 | 0.887 |
| DiffusionMBIR | 25.98 | 0.872 | 27.14 | 0.877 | 27.74 | 0.903 | 30.52 | 0.906 | 30.57 | 0.906 | 30.68 | 0.907 |
| TPDM | - | - | - | - | - | - | 14.44 | 0.141 | 14.06 | 0.141 | 14.54 | 0.313 |
| DDS 2D | 28.05 | 0.916 | 27.99 | 0.916 | 28.82 | 0.922 | 27.92 | 0.843 | 27.89 | 0.835 | 27.96 | 0.842 |
| DDS | 28.20 | 0.918 | 28.17 | 0.926 | 29.03 | 0.934 | 28.12 | 0.865 | 28.06 | 0.869 | 28.13 | 0.879 |
| DiffusionBlend (Ours) | 35.38 | 0.971 | 35.85 | 0.972 | **37.62** | 0.972 | 30.43 | 0.917 | 31.24 | 0.920 | 31.02 | 0.924 |
| DiffusionBlend++ (Ours) | **35.86** | **0.975** | **36.03** | **0.976** | 37.45 | **0.976** | **34.33** | **0.957** | **34.48** | **0.957** | **34.64** | **0.956** |

Table 4: Comprehensive comparison of quantitative results on Limited-Angle CT Reconstruction on All Views for AAPM and LIDC datasets. Best results are in bold.

run. FBP-UNet performed reasonably well, but is a supervised method where the network must be retrained for each specific task. DDS is the most directly comparable to our method in runtime and methodology, but performed much worse quantitatively.

**Inter-slice smoothness** We demonstrate that DiffusionBlend++ learns the 3D prior internally, and achieves consistency and smoothness between 2D axial-plane slices without any external regularizations. In Table 5, we present the total variation (TV) value of the reconstructed images of different reconstruction algorithms on the test set of AAPM dataset, given by

| Algorithm | TV value | Difference with gt |
|---|---|---|
| DDS 2D | 0.0104 | 0.0044 |
| DDS | 0.0031 | -0.0034 |
| DiffusionBlend++ (Ours) | 0.0043 | **-0.0022** |
| Ground Truth | 0.0065 | - |

Table 5: TV values of different reconstruction algorithms on the AAPM test set

$\frac{1}{C \times W \times H}||\mathbf{D}_z(x)||_1$, where $x$ is the image, $\mathbf{D}_z$ is the total variation operator in $z$ direction, and $C$, $W$, $H$ are number of channels, width, and height. We find that both DiffusionBlend++ and DDS have TV less than the ground truth image, which implies that the reconstructed images are smooth in the z direction. However, we observe that DDS over-smooths the images as demonstrated in Fig. 5, which is represented by a much lower TV value than the ground truth. On the other hand, DiffusionBlend++ has smoothness level close to the ground truth without sacrificing sharpness of images.

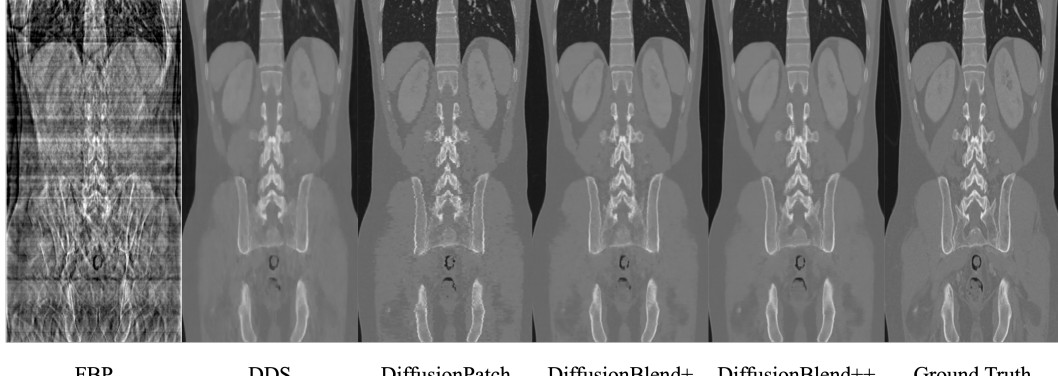

| FBP | DDS | DiffusionPatch | DiffusionBlend+ | DiffusionBlend++ | Ground Truth |

Figure 5: Results of CT reconstruction with 8 views on AAPM dataset, coronal view. DiffusionPatch refers to Algorithm 1 with the same partition for every timestsep, and DiffusionBlend+ refers to Algorithm 1 only with partitions of adjacency slices.

**Effectiveness of adjacency-slice blending and cross-slice blending**  We demonstrate that both the adjacency-slice blending and the cross-slice blending module are instrumental to a better reconstruction quality. Table 7 demonstrates the effectiveness of adding blending modules to the reverse sampling. Given the pretrained diffusion prior over slice patches, we observe that adding the adjacency-slice blending module improves the PSNR over a fixed partition by 1.17dB, and adding an additional cross-slice blending module further improves the PSNR by 1.63dB. Fig. 5 demonstrates that adding the cross-slice blending module removes artifacts and recovers sharper edges.

**Ablation Studies** We investigated the performance gain due to individual components. Details can be found in Appendix A.3.

| Adjacency | Cross | PSNR ↑ | SSIM ↑ |
|-----------|-------|--------|--------|
|           |       | 34.85  | 0.954  |
| ✓         |       | 36.02  | 0.965  |
| ✓         | ✓     | **36.48** | **0.968** |

Table 6: Effectiveness of Blending Modules, Sagittal view performance on AAPM

## 5  Conclusion

In this work, we proposed two methods of using score-based diffusion models to learn priors of three dimensional volumes and used them to perform CT reconstruction. In both cases, we learn the distributions of multiple slices of a volume at once and blend the distributions together at inference time. Extensive experiments showed that our method substantially outperformed existing methods for 3D CT reconstruction both quantitatively and qualitatively in the sparse view and limited angle settings. In the future, more work could be done on other 3D inverse problems and acceleration through latent diffusion models. Image reconstruction methods like those proposed in this paper have the potential to benefit society by reducing X-ray dose in CT scans.

## Acknowledgments and Disclosure of Funding

The authors acknowledge support from Michigan Institute for Computational Discovery and Engineering (MICDE) Catalyst Grant, and Michigan Institute for Data Science (MIDAS) PODS Grant.

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

# A Appendix / supplemental material

## A.1 Score matching derivations for DiffusionBlend++

We show how the score matching method described in [1] can be simplified in the case of assumptions such as the ones described in Table 12.

**Product of distributions.** Suppose first that the distribution of interest can be expressed as $p(\boldsymbol{x}) = q(\boldsymbol{x})^a r(\boldsymbol{x})^b / Z$ for density functions $q$ and $r$, constant positive scalars $a$ and $b$, and a scaling factor $Z$. Following [1], to learn the score function, we can minimize the loss function

$$\mathbb{E}_{t \sim \mathcal{U}(0,T)} \mathbb{E}_{\boldsymbol{x} \sim p(\boldsymbol{x})} \mathbb{E}_{\boldsymbol{y} \sim \mathcal{N}(\boldsymbol{x}, \sigma_t^2 I)} \| \boldsymbol{s}_\theta(\boldsymbol{y}, \sigma_t) - \frac{\boldsymbol{y} - \boldsymbol{x}}{\sigma_t^2} \|_2^2, \tag{9}$$

where $\boldsymbol{s}_\theta$ represents a neural network. Denoting the score functions of $p, q$, and $r$ by $\boldsymbol{s}, \boldsymbol{s}_p$, and $\boldsymbol{s}_q$, we have $\boldsymbol{s}(\boldsymbol{x}) = a \boldsymbol{s}_q(\boldsymbol{x}) + b \boldsymbol{s}_r(\boldsymbol{x})$. Hence, if we instead use neural networks to learn $\boldsymbol{s}_p$ and $\boldsymbol{s}_q$, we could minimize the loss function

$$L_1 = \mathbb{E}_{t \sim \mathcal{U}(0,T)} \mathbb{E}_{\boldsymbol{x} \sim p(\boldsymbol{x})} \mathbb{E}_{\boldsymbol{y} \sim \mathcal{N}(\boldsymbol{x}, \sigma_t^2 I)} \| a \boldsymbol{s}_{q,\theta}(\boldsymbol{y}, \sigma_t) + b \boldsymbol{s}_{r,\theta}(\boldsymbol{y}, \sigma_t) - \frac{\boldsymbol{y} - \boldsymbol{x}}{\sigma_t^2} \|_2^2. \tag{10}$$

However, this loss function is computationally expensive to work with, as backpropagation through both networks is necessary. Thus, it would be ideal to derive a simpler form of this loss function.

Toward these ends, for simplicity we define $X = a \boldsymbol{s}_{q,\theta}(\boldsymbol{y}, \sigma_t) - \frac{a}{a+b} \cdot \frac{\boldsymbol{y} - \boldsymbol{x}}{\sigma_t^2}$ and $Y = b \boldsymbol{s}_{r,\theta}(\boldsymbol{y}, \sigma_t) - \frac{b}{a+b} \cdot \frac{\boldsymbol{y} - \boldsymbol{x}}{\sigma_t^2}$, where all images have been vectorized. Now

$$\| X - Y \|_2^2 = \| X \|_2^2 + \| Y \|_2^2 - 2 \langle X, Y \rangle \geq 0. \tag{11}$$

Thus, rearranging the inequality and adding $\| X \|_2^2 + \| Y \|_2^2$ to both sides yields $\| X + Y \|_2^2 \leq 2 \| X \|_2^2 + 2 \| Y \|_2^2$.

Returning to the original loss function, we have

$$L_1 = \mathbb{E}_{t \sim \mathcal{U}(0,T)} \mathbb{E}_{\boldsymbol{x} \sim p(\boldsymbol{x})} \mathbb{E}_{\boldsymbol{y} \sim \mathcal{N}(\boldsymbol{x}, \sigma_t^2 I)} \| X + Y \|_2^2. \tag{12}$$

By applying the inequality proven above, we get

$$L_1 \leq 2 \mathbb{E}_{t \sim \mathcal{U}(0,T)} \mathbb{E}_{\boldsymbol{x} \sim p(\boldsymbol{x})} \mathbb{E}_{\boldsymbol{y} \sim \mathcal{N}(\boldsymbol{x}, \sigma_t^2 I)} \| a \cdot \boldsymbol{s}_{q,\theta}(\boldsymbol{y}, \sigma_t) - \frac{a}{a+b} \cdot \frac{\boldsymbol{y} - \boldsymbol{x}}{\sigma_t^2} \|_2^2 \tag{13}$$

$$+ 2 \mathbb{E}_{t \sim \mathcal{U}(0,T)} \mathbb{E}_{\boldsymbol{x} \sim p(\boldsymbol{x})} \mathbb{E}_{\boldsymbol{y} \sim \mathcal{N}(\boldsymbol{x}, \sigma_t^2 I)} \| b \cdot \boldsymbol{s}_{r,\theta}(\boldsymbol{y}, \sigma_t) - \frac{b}{a+b} \cdot \frac{\boldsymbol{y} - \boldsymbol{x}}{\sigma_t^2} \|_2^2. \tag{14}$$

For the special case of $a = b = \frac{1}{2}$, this inequality is rewritten as

$$L_1 \leq \mathbb{E}_{t \sim \mathcal{U}(0,T)} \mathbb{E}_{\boldsymbol{x} \sim p(\boldsymbol{x})} \mathbb{E}_{\boldsymbol{y} \sim \mathcal{N}(\boldsymbol{x}, \sigma_t^2 I)} \| \boldsymbol{s}_{q,\theta}(\boldsymbol{y}, \sigma_t) - \frac{\boldsymbol{y} - \boldsymbol{x}}{\sigma_t^2} \|_2^2 \tag{15}$$

$$+ \mathbb{E}_{t \sim \mathcal{U}(0,T)} \mathbb{E}_{\boldsymbol{x} \sim p(\boldsymbol{x})} \mathbb{E}_{\boldsymbol{y} \sim \mathcal{N}(\boldsymbol{x}, \sigma_t^2 I)} \| \boldsymbol{s}_{r,\theta}(\boldsymbol{y}, \sigma_t) - \frac{\boldsymbol{y} - \boldsymbol{x}}{\sigma_t^2} \|_2^2. \tag{16}$$

Note that each of the two individual terms in the sum precisely represents the score matching equation for learning the score functions $\boldsymbol{s}_p$ and $\boldsymbol{s}_q$. Hence, to train the networks $\boldsymbol{s}_{q,\theta}$ and $\boldsymbol{s}_{r,\theta}$ by minimizing $L_1$, we may instead minimize the upper bound of $L_1$ by separately training these two networks.

In practice, we may opt to use the same network for $\boldsymbol{s}_{q,\theta}$ and $\boldsymbol{s}_{r,\theta}$ but with an additional input specifying which distribution between $q$ and $r$ to use. In this case, at each training iteration, we randomly choose from one of the two distributions and perform backpropagation using this distribution. More precisely, we redefine our network $\boldsymbol{s}_\theta(\boldsymbol{x}, \sigma_t, v)$ with $v$ being either 0 or 1. When $v = 0$, $\boldsymbol{s}_\theta(\boldsymbol{x}, \sigma_t, v) = \boldsymbol{s}_{q,\theta}(\boldsymbol{x}, \sigma_t)$ and when $v = 1$, $\boldsymbol{s}_\theta(\boldsymbol{x}, \sigma_t, v) = \boldsymbol{s}_{r,\theta}(\boldsymbol{x}, \sigma_t)$. Thus the loss bound becomes

$$L_1 \leq \mathbb{E}_{t \sim \mathcal{U}(0,T)} \mathbb{E}_{\boldsymbol{x} \sim p(\boldsymbol{x})} \mathbb{E}_{\boldsymbol{y} \sim \mathcal{N}(\boldsymbol{x}, \sigma_t^2 I)} \mathbb{E}_{v \in \{0,1\}} \| \boldsymbol{s}_\theta(\boldsymbol{y}, \sigma_t, v) - \frac{\boldsymbol{y} - \boldsymbol{x}}{\sigma_t^2} \|_2^2. \tag{17}$$

Finally, this derivation easily extends to the more general case where the distribution of interest can be expressed as

$$p(\boldsymbol{x}) = \prod_{i=1}^{k} p_i(\boldsymbol{x})^{1/k}/Z. \tag{18}$$

In this case, the similarly defined score matching loss function $L_1$ can be upper bounded by an expression similar to (17), but with $v$ being randomly selected from $k$ possible values.

In summary, we have shown that for the case of a decomposable distribution $p(\boldsymbol{x})$, the score function of $p(\boldsymbol{x})$ can be learned simply through the score function of the individual components $p_i(\boldsymbol{x})$. In the special case when each of the components have equal weight, it suffices to randomly choose one of the components and backpropagate through the score matching loss function according to that component.

**Separable distributions.** Next, we show how the score matching method is simplified for distributions of the form $p(\boldsymbol{x}) = \prod_{i=1}^{r} p(\boldsymbol{x}[:,:,\mathcal{S}_i])/Z$, where the same notation as Table 12 is used and $\mathcal{S} = \mathcal{S}_1 \cup \ldots \cup \mathcal{S}_r$ denotes an arbitrary partition of $\{1, 2, \ldots, H\}$. The score function of $p(\boldsymbol{x})$ can be written as

$$s(\boldsymbol{x}) = \sum_{i=1}^{H} \nabla \log p(\boldsymbol{x}[:,:,\mathcal{S}_i]) = \sum_{i=1}^{H} \boldsymbol{s}_i(\boldsymbol{x}[:,:,\mathcal{S}_i]), \tag{19}$$

where $\boldsymbol{s}_i$ represents the score function of the slices of $\boldsymbol{x}$ corresponding to $\mathcal{S}_i$. Then (9) becomes

$$L = \mathbb{E}_{t\sim\mathcal{U}(0,T)}\mathbb{E}_{\boldsymbol{x}\sim p(\boldsymbol{x})}\mathbb{E}_{\boldsymbol{y}\sim\mathcal{N}(\boldsymbol{x},\sigma_t^2 I)} \left\| \sum_{i=1}^{H} \boldsymbol{s}_{\theta,i}(\boldsymbol{x}[:,:,\mathcal{S}_i]) - \frac{\boldsymbol{y}-\boldsymbol{x}}{\sigma_t^2} \right\|_2^2. \tag{20}$$

Since each of the $\mathcal{S}_i$'s are disjoint, this can be broken up and rewritten as

$$L = \sum_{i=1}^{H} \mathbb{E}_{t\sim\mathcal{U}(0,T)}\mathbb{E}_{\boldsymbol{x}\sim p(\boldsymbol{x})}\mathbb{E}_{\boldsymbol{y}\sim\mathcal{N}(\boldsymbol{x},\sigma_t^2 I)} \left\| \boldsymbol{s}_{\theta,i}(\boldsymbol{x}[:,:,\mathcal{S}_i]) - \frac{\boldsymbol{y}[:,:,\mathcal{S}_i] - \boldsymbol{x}[:,:,\mathcal{S}_i]}{\sigma_t^2} \right\|_2^2. \tag{21}$$

Thus, after replacing the outer sum with an expectation over $i$, this is equivalent to randomly choosing one of the partitions $\mathcal{S}_i$ and performing denoising score matching on only $\boldsymbol{x}[:,:,\mathcal{S}_i]$.

A very similar derivation holds for the general case where the 3D volume $\boldsymbol{x}$ can be partitioned into an arbitrary number of smaller volumes of any shape $\boldsymbol{x} = \boldsymbol{x}_1 \cup \boldsymbol{x}_2 \cup \ldots \cup \boldsymbol{x}_H$ and $p(\boldsymbol{x}) = \prod_{i=1}^{H} p(\boldsymbol{x}_i)/Z$. For this case, training consists of randomly selecting one of the partitions at each iteration and performing score matching on that partition. For example, when $\boldsymbol{x}_i = \boldsymbol{x}[:,:,i]$, it is common to select 2D slices from the training volumes and learn a two dimensional diffusion model on those slices [13, 14].

**Applying to DiffusionBlend++.** When $p(\boldsymbol{x})$ follows the distribution in DiffusionBlend++ we can combine the results of the previous two sections to show how to perform score matching. In the first part of this section, we showed how to perform score matching for a distribution expressed as a product of "simpler" distributions by performing score matching on the individual distributions. DiffusionBlend++ follows this assumption where

$$p_i(\boldsymbol{x}) = \left( \prod_{j=1}^{r} p(\boldsymbol{x}[:,:,\mathcal{S}_j]) \right) /Z_i. \tag{22}$$

Here, $i$ represents an index that can iterate through the ways of partitioning $\mathcal{S} = \mathcal{S}_1 \cup \ldots \cup \mathcal{S}_r$. The input $v$ to the network specifying which of the simpler distributions is used is embedded as the relative position encoding for each of the partitions as described in Section 3. Finally, to learn the score function of $p_i(\boldsymbol{x})$, we can use the loss function in (21).

## A.2 Additional Algorithms

The reconstruction algorithm for DiffusionBlend is provided below.

The training algorithms for DiffusionBlend and DiffusionBlend++ are provided below.

**Algorithm 2** DiffusionBlend

**Require:** $\boldsymbol{A}$, $M$, $\zeta_i > 0$, $j$, $\boldsymbol{y}$
 Initialize $\boldsymbol{x}_T \sim \mathcal{N}(0, \sigma_T^2 \boldsymbol{I})$
 **for** $t = T : 1$ **do**
  For each $i$ compute $\boldsymbol{\epsilon}_\theta(\boldsymbol{x}_t[:,:,i]|\boldsymbol{x}_t[:,:,i-j:i-1], \boldsymbol{x}_t[:,:,i+1:i+j])$
  Compute $\boldsymbol{s} = \nabla \log p(\boldsymbol{x}_t)$ using (4)
  Compute $\hat{\boldsymbol{x}}_t = \mathbb{E}[\boldsymbol{x}_0|\boldsymbol{x}_t]$ using Tweedie's formula
  Set $\hat{\boldsymbol{x}}_t' = \mathrm{CG}(\boldsymbol{A}^*\boldsymbol{A}, \boldsymbol{A}^*\boldsymbol{y}, \hat{\boldsymbol{x}}_t)$
  Sample $\boldsymbol{x}_{t-1}$ using $\hat{\boldsymbol{x}}_t'$ and $\boldsymbol{s}$ via DDIM sampling
 **end for**
Return $\boldsymbol{x}$.

---

**Algorithm 3** DiffusionBlend training

 **repeat**
 Select $\boldsymbol{x} \sim p(\boldsymbol{x})$
 Select $t \sim \mathrm{Uniform}[1, T]$
 Set $\epsilon \sim \mathcal{N}(0, I)$
 Select $i \sim \mathrm{Uniform}[1, H]$
 Take gradient descent step on $\nabla_\theta \|(\epsilon_\theta(\mathbf{x}_t[:,:,i-j:i+j], \sigma_t) - \epsilon[:,:,i])\|_2^2$
 **until converged**
Return $D_\theta$

---

**Algorithm 4** DiffusionBlend++ training

 **repeat**
 Select $\boldsymbol{x} \sim p(\boldsymbol{x})$
 Select $t \sim \mathrm{Uniform}[1, T]$
 Set $\epsilon \sim \mathcal{N}(0, I)$
 Select a partition $\mathcal{S} = \mathcal{S}_1 \cup \ldots \cup \mathcal{S}_r$
 Select $i \sim \mathrm{Uniform}[1, r]$
 Take gradient descent step on $\nabla_\theta \|(\epsilon_\theta(\mathbf{x}_t[:,:,\mathcal{S}_i], \sigma_t) - \epsilon[:,:,\mathcal{S}_i])\|_2^2$
 **until converged**
Return $D_\theta$

## A.3 Ablation studies

We run the following ablation studies to examine each of the individual components of our DiffusionBlend++ method. Firstly, we examine the performance gain of adding adjacency slicue blending (DiffusionBlend+) and adding cross-slice blending. Next, we examine the effect of including the positional encoding as an input to the network. Then we look at the quantitative metrics

| Adjacency | Cross | PSNR ↑ | SSIM ↑ |
|-----------|-------|--------|--------|
|           |       | 34.85  | 0.954  |
| ✓         |       | 36.02  | 0.965  |
| ✓         | ✓     | **36.48** | **0.968** |

Table 7: Effectiveness of Blending Modules, Sagittal view performance on AAPM

of the reconstructed images when applying different numbers of NFEs for the comparison methods. Finally, we examine the effect of choosing different slices for each partition.

**Effectiveness of adjacency-slice blending and cross-slice blending**   We demonstrate that both the adjacency-slice blending and the cross-slice blending module are instrumental to a better reconstruction quality. Table 7 demonstrates the effectiveness of adding blending modules to the reverse sampling. Given the pretrained diffusion prior over slice patches, we observe that adding the adjacency-slice blending module improves the PSNR over a fixed partition by 1.17dB, and adding an additional cross-slice blending module further improves the PSNR by 1.63dB. Fig. 5 demonstrates that adding the cross-slice blending module removes artifacts and recovers sharper edges.

**Robust performance with low NFEs.**   Since DDS and DiffusionBlend++ both use the DDIM sampler for acceleration, we performed experiments with both of these methods using different NFEs. The left of Fig. 6 shows graphs of these two methods and Table 8 shows the quantitative results. DDS is very sensitive to the number of NFEs used and there is a sharp dropoff in PSNR if too few or too many NFEs are used. On the other hand, DiffusionBlend++ performs the best for the highest number of NFEs due to the slice blending strategy while still obtaining superior results for 50 NFEs. Also, this method is much more robust to varying NFEs, displaying only 1.4dB of variance in the shown results compared to 2.4dB of varaince for DDS. For fair comparisons, we use 200 NFEs for all the main experiments.

**Frequency of applying slice jumps.**   To demonstrate the use of jump slice partitions at reconstruction time, we performed experiments varying the frequency of applying these jump slices. For instance, for a frequency of 8, the reconstruction algorithm consisted of updating the volume using jump slices for iteration numbers that are a multiple of 8 and updating using adjacent slices for all other iterations. The right of Fig. 6 shows a graph of the results for different frequencies and the

Table 8: Axial PSNR for 8 view SVCT recon for different NFEs

| Method | 50 | 100 | 200 | 334 |
|--------|-----|------|------|------|
| DDS | 30.8 | 32.2 | **33.2** | 32.7 |
| DiffusionBlend++ | 34.5 | 35.0 | 35.7 | **35.9** |

quantitative results are presented in Table 9. The best results are obtained when the frequency is 2, corresponding to alternating updates with adjacent slices and jump slices, and the PSNR decreases monotonically as the frequency increases. This indicates that the jump slices capture more nonlocal information across a volume and help to improve the image quality.

Table 9: Axial PSNR for 8 view SVCT recon for different slice jump frequencies

| Frequency | 2 | 4 | 8 | 12 | 16 | 32 |
|-----------|-----|------|------|------|------|------|
| PSNR | **35.69** | 35.62 | 35.50 | 35.45 | 35.37 | 35.28 |

## A.4 Additional Results

**Classical Baselines and more Projection Angles**   We provide additional results with classical baselines (without deep learning) such as SIRT, SBTV, and CGLS for the LDCT dataset. Results show that our method outperforms the baselines significantly for every angle we evaluated on. DiffusionBlend++ starts to reconstruct images very close to the ground truth with 20 projections or more, but other baselines such as SIRT and CGLS still struggle to get a satisfying reconstruction with

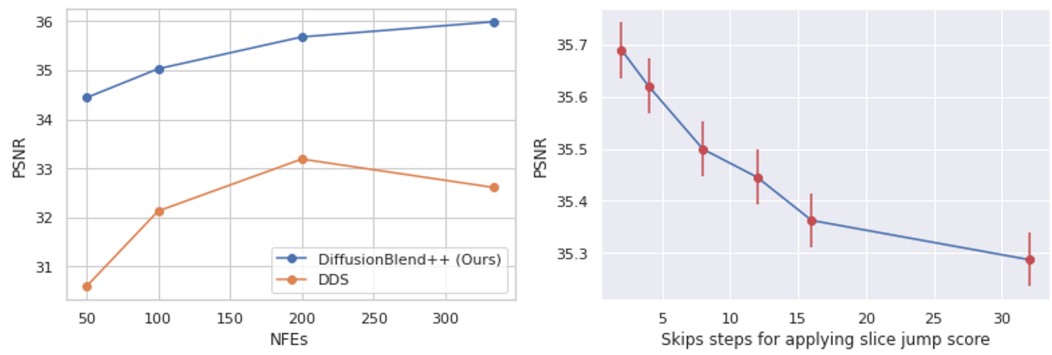

Figure 6: Quantitative results (axial view) of CT reconstruction with 8 views on AAPM dataset for different NFEs and slice blending methods.

Table 10: Wall times of various methods for 8 view 3D CT reconstruction

| Method | NFEs | Wall time (min) |
| --- | --- | --- |
| DiffusionMBIR | 2000 | 1400 |
| TPDM | 2000 | 1200 |
| DDS | 200 | 48 |
| DiffusionBlend | 200 | 70 |
| DiffusionBlend++ | 200 | **32** |

60 views or more. Fig. 7 shows the reconstruction performance on the coronal plane of different methods. We observe that DiffusionBlend++ (ours) has a significant margin above baselines for every view. Note that DiffusionBlend++ still outperforms DDS2D significantly with >40 views, which demonstrates that our 3D prior is still very useful even with much more views. We also simulate low-dose noise to the reconstruction, which showing our algorithm is robust to noise by a minor decrease in reconstruction performance. Our method (DiffusionBlend++) is shown to outperforms all baselines at every angle as in Fig. 8.

**Error Bars** We demonstrate the standard deviation of the results with sparse-view CT reconstruction on AAPM and LIDC dataset here to demonstrate that the result is statistically sigificant.

Fig. 9 shows the visual results for SVCT reconstruction with 8 views on the LIDC dataset.

Fig. 10 shows the visual results for SVCT reconstruction with 6 views on the LIDC dataset.

Fig. 11 shows the visual results for SVCT reconstruction with 4 views on the LIDC dataset.

Fig. 12 shows the visual results for LACT reconstruction on the LIDC dataset.

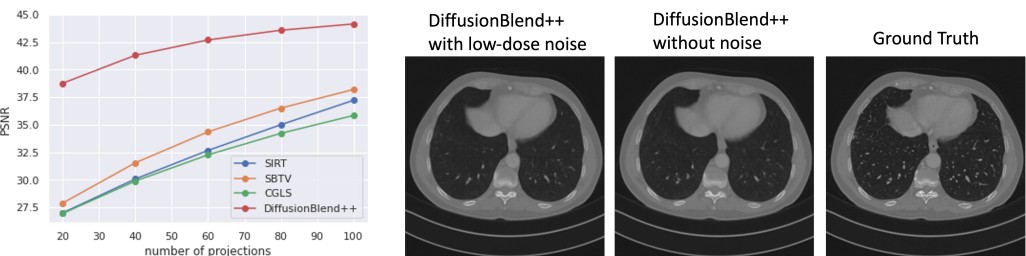

Figure 7: Left: Performance of DiffusionBlend++ on more angles, Right: Reconstruction of DiffusionBlend++ with low-dose noise

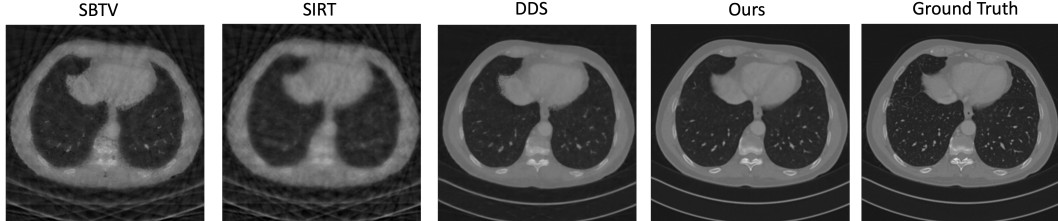

Figure 8: Comparison of DiffusionBlend++ with classical methods

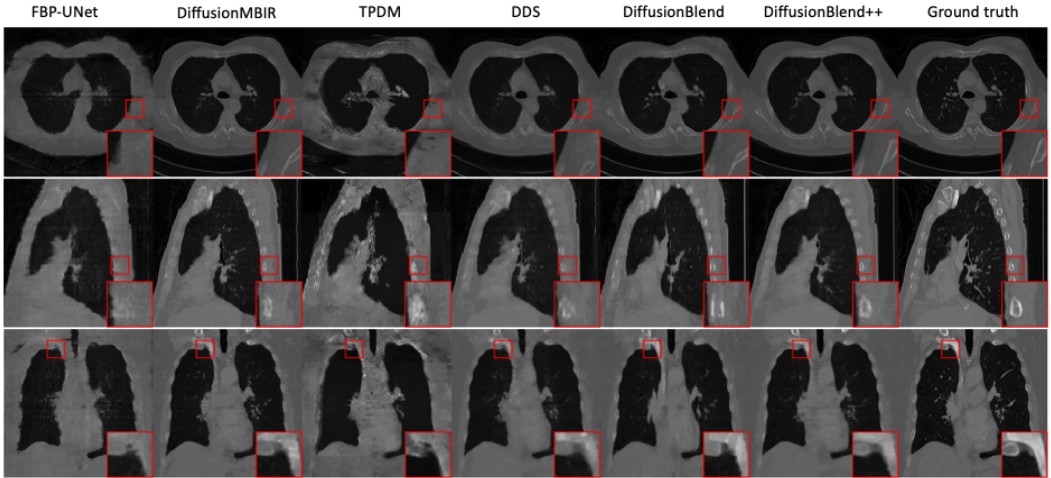

Figure 9: Results of 3D CT reconstruction with 8 views on LIDC dataset. Top row is axial view, middle row is sagittal view, bottom row is coronal view.

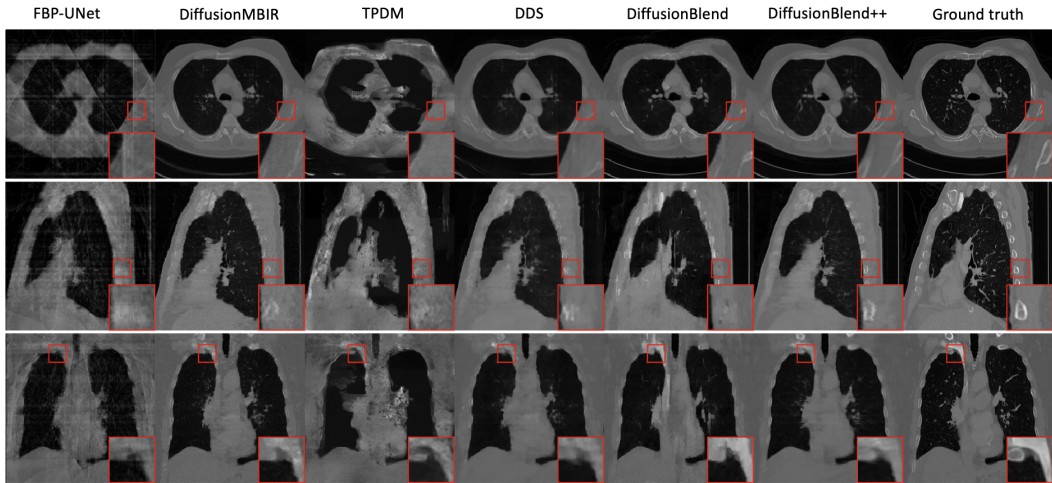

Figure 10: Results of 3D CT reconstruction with 6 views on LIDC dataset. Top row is axial view, middle row is sagittal view, bottom row is coronal view.

| Method | Sparse-View CT Reconstruction on AAPM | | | | | | Sparse-View CT Reconstruction on LIDC | | | | | |
|---|---|---|---|---|---|---|---|---|---|---|---|---|
| | 8 views | | 6 views | | 4 views | | 8 Views | | 6 Views | | 4 Views | |
| | PSNR↑ | SSIM↑ | PSNR↑ | SSIM↑ | PSNR↑ | SSIM↑ | PSNR↑ | SSIM↑ | PSNR↑ | SSIM↑ | PSNR↑ | SSIM↑ |
| FBP | 2.64 | 0.16 | 2.88 | 0.26 | 2.91 | 0.20 | 2.57 | 0.17 | 2.78 | 0.21 | 2.84 | 0.16 |
| FBP-UNet | 3.46 | 0.30 | 3.34 | 0.28 | 3.04 | 0.31 | 3.42 | 0.31 | 3.23 | 0.29 | 3.14 | 0.27 |
| DiffusionMBIR | 1.93 | 0.08 | 1.48 | 0.13 | 1.28 | 0.11 | 1.89 | 0.09 | 1.56 | 0.12 | 1.31 | 0.14 |
| TPDM | - | - | - | - | - | - | 2.32 | 0.14 | 2.02 | 0.16 | 2.52 | 0.19 |
| DDS 2D | 1.76 | 0.07 | 2.04 | 0.10 | 2.73 | 0.11 | 1.85 | 0.09 | 2.13 | 0.13 | 267 | 0.18 |
| DDS 2D | 1.76 | 0.07 | 2.04 | 0.10 | 2.73 | 0.11 | 1.85 | 0.09 | 2.13 | 0.13 | 2.67 | 0.18 |
| DDS 2D | 1.68 | 0.06 | 1.96 | 0.09 | 2.65 | 0.11 | 1.84 | 0.09 | 2.10 | 0.12 | 2.64 | 0.18 |
| DiffusionBlend (Ours) | 1.67 | 0.06 | 1.78 | 0.08 | 1.98 | 0.09 | 1.70 | 0.07 | 2.03 | 0.11 | 2.54 | 0.16 |
| DiffusionBlend++ (Ours) | 1.50 | **0.06** | 1.65 | 0.08 | 1.71 | 0.10 | 1.60 | 0.09 | 1.68 | 0.10 | 1.82 | 0.11 |

Table 11: Standard Deviation of Performance on Sparse-View CT Reconstruction on Sagittal View for AAPM and LIDC datasets. Best results are in bold.

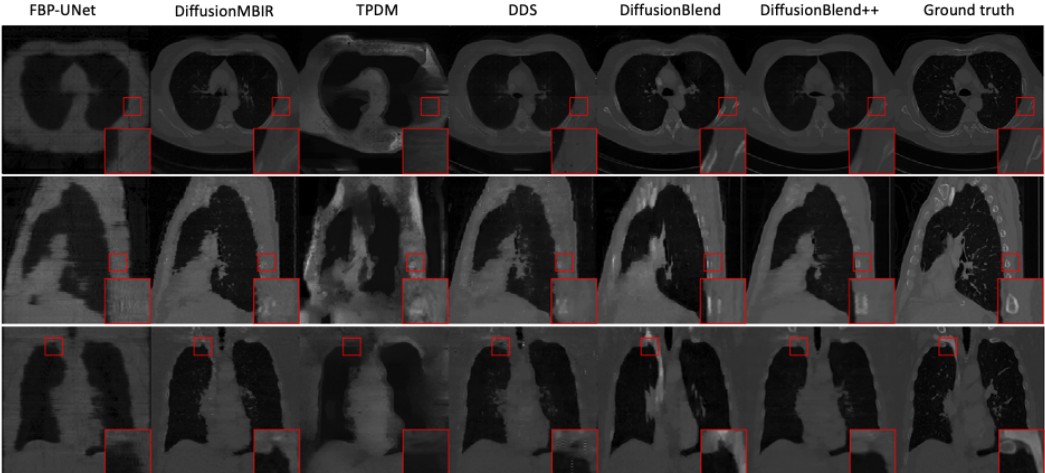

Figure 11: Results of 3D CT reconstruction with 4 views on LIDC dataset. Top row is axial view, middle row is sagittal view, bottom row is coronal view.

## A.5 Experiment parameters

Since axial slices belonging to the same volume that are far apart have limited correlation, DiffusionBlend++ selects only partitions of $\mathcal{S}$ for training where slices belonging to the same partition are fairly close to one another. Then the same range of possible partition schemes are used during reconstruction time. More precisely, we take the size of each $\mathcal{S}_i$ to be 3 and first repetition pad the volume so that the number of axial slices is a multiple of 9. Then we consider the following partitions:

- $\mathcal{S}_1 = \{1, 2, 3\}$, $\mathcal{S}_2 = \{4, 5, 6\}$, $\mathcal{S}_3 = \{7, 8, 9\}$. Furthermore, for all integers $k > 1$, $\mathcal{S}_k = \mathcal{S}_{k-3} \bigoplus 9\lfloor (k-1)/3 \rfloor$, where $\bigoplus$ represents adding the same number to each element of the set. For example, $\mathcal{S}_4 = \{10, 11, 12\}$, $\mathcal{S}_5 = \{13, 14, 15\}$, $\mathcal{S}_6 = \{16, 17, 18\}$.

- $\mathcal{S}_1 = \{1, 4, 7\}$, $\mathcal{S}_2 = \{2, 5, 8\}$, $\mathcal{S}_3 = \{3, 6, 9\}$. Furthermore, for all integers $k > 1$, $\mathcal{S}_k = \mathcal{S}_{k-3} \bigoplus 9\lfloor (k-1)/3 \rfloor$.

Table 12: 3D prior modeling methods

| Method | Distribution Model |
|---|---|
| DiffusionMBIR [13] | $\prod_{i=1}^{H} p(\boldsymbol{x}[:,:,i])/Z$ |
| TPDM [14] | $\left( \prod_{i=1}^{N} q_\theta(\boldsymbol{x}[:,:,i])^\alpha \right) \left( \prod_{j=1}^{N} q_\phi(\boldsymbol{x}[j,:,:])^\beta \right) /Z$ |
| DiffusionBlend | $\prod_{i=1}^{H} p(\boldsymbol{x}[:,:,i] \| \boldsymbol{x}[:,:,i-j:i-1], \boldsymbol{x}[:,:,i+1:i+j])/Z$ |
| DiffusionBlend++ | $\prod_{i=1}^{r} p(\boldsymbol{x}[:,:,\mathcal{S}_i])/Z$ |

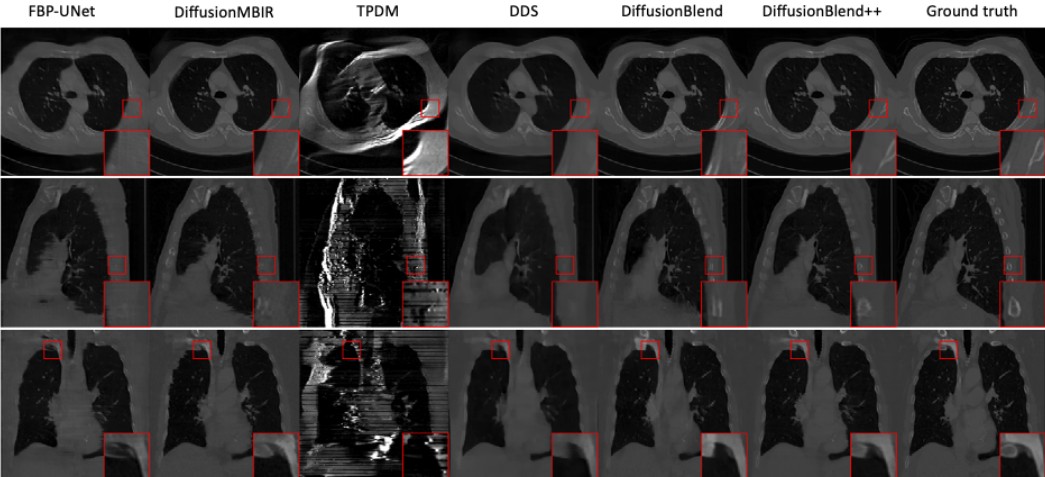

Figure 12: Results of limited angle 3D CT reconstruction on LIDC dataset. Top row is axial view, middle row is sagittal view, bottom row is coronal view.

## A.6 Comparison experiment details

**FBP-UNet.** We used the same neural network architecture as the original paper [57]. Individual networks were trained for each of the 8 view, 6 view, 4 view, and LACT experiments for each of the datasets. Each of the networks were trained from scratch with a batch size of 32 for 150 epochs.

**DiffusionMBIR.** We separately trained networks for the AAPM and LIDC datasets by fine-tuning the original checkpoint provided in [13] for 100 and 10 epochs, respectively. The batch size was set to 4. For reconstruction, we used the same set of hyperparameters for all of the experiments: $\lambda = 0.04$, $\rho = 10$, and $r = 0.16$ for the sampling algorithm. 2000 NFEs were used for the diffusion process.

**TPDM.** We fine-tuned the axial and sagittal checkpoints provided in [14] on the LIDC dataset for 10 epochs. For reconstruction, we used 2000 NFEs and alternated between updating the volume using the axial checkpoint and sagittal checkpoint, with each checkpoint being used equally frequently. The DPS step size parameter was set to $\zeta = 0.5$.

**DDS.** We separately trained networks for the AAPM and LIDC datasets by fine-tuning the original checkpoint provided in [15] for 100 and 10 epochs, respectively. We used 100 NFEs at reconstruction as this was observed to give the best performance. The reconstruction parameters were set to $\eta = 0.85$, $\lambda = 0.4$, and $\rho = 10$. Five iterations of conjugate gradient descent were run per diffusion step. For DDS 2D, the parameters were left unchanged with the exception of using $\rho = 0$ to avoid enforcing the TV regularizer between slices.

**SBTV.** We implement this algorithm with variables splitting of 3D anisotropic TV regularization (Dz, Dx, and Dy). We first check number of iterations, note that the performance converges with around 30 iterations. We did a grid search of hyperparameters on 9 validation images (not in the test set) for every projection angles.

**SIRT.** This algorithm iteratively updates the reconstruction based on the residual between projection of the reconstruction and the GT. It only has the number of iterations as its hyper-parameter. We note that during inference, PSNR increases with more iterations, but saturates later. So we set the total number iterations to be 1000, with an early stopping threshold of 1e-6 between two consecutive iterations.

**CGLS.** This algorithm uses conjugate gradient for solving least square problems. In our case, we use $CG(A^T A + \rho x^T x, A^T y)$, $\rho$ is set to be 1e-4 based on grid search for numerical stability. We tune the number of iterations on validation set, and find that performance saturates at around 25 iterations.

## A.7 Limitations

One limitation of our work is that we use noiseless simulated measurements for all our experiments. The robustness of our method to noise added to the measurements should be explored further. Likewise, future work should evaluate the accuracy of our method when applied to real measurement data, which will contain measurement noise and mismatches between the true system model and used forward model. Another limitation of our work is a lack of other types of 3D image reconstruction applications shown. Although the proposed method is unsupervised and the reconstruction algorithm can be readily be applied to other 3D linear inverse problems, future work should explore other applications of DiffusionBlend.

