# OpenReview forum: "DiffusionBlend: Learning 3D Image Prior through Position-aware Diffusion Score Blending for 3D Computed Tomography Reconstruction"
_NeurIPS.cc/2024/Conference — NeurIPS 2024 poster_

### Official Review · Reviewer_FN1q · 2024-07-09

**Soundness:** 3
**Presentation:** 3
**Contribution:** 2
**Rating:** 7
**Confidence:** 5

**Summary:**

This paper used the diffusion model to solve the widely known inverse problem in computed tomography. The paper is well-written and well-organized.
I reckon this paper has two main contributions:
1. The first diffusion model in CT considered z-axis consistency and used 3d data as neural network input.
2. A SOTA results with astonishing visualization and metrics results.

**Strengths:**

1. The first diffusion model in CT considered z-axis consistency and used 3d data as neural network input.
2. A SOTA results with astonishing visualization and metrics results.

**Weaknesses:**

1. I admit this is the first paper I read using the diffusion model while considering z-axis consistency. However, the y-axis has been widely discussed and solved using various types of regularization between 2D slices or just using 3D data before. I consider the author just used a 3D way here to solve this point. Yeah, it is new for the diffusion model, but I do think it is less attractive to me considering the works already made before.
2. The results achieved by the authors, particularly in sparse-view reconstruction using only four angles, are indeed remarkable. The quality of the reconstruction with such limited data is surprising and commendable. Given the significance of these results, it would be beneficial for the community if the authors consider sharing their code upon acceptance to enable further research and validation.
3. The paper's novelty is somewhat overshadowed by the work "Solving Inverse Problems in Medical Imaging with Score-Based Generative Models," which the authors have cited. It would be beneficial for the authors to clearly state the innovations and contributions of their work, especially in light of the foundational problem settings of inverse problem solving for medical imaging already addressed by the cited paper.

**Questions:**

1. What is your window center and width for CT images in the main body of the paper? I think the current window width value you selected for the main part of the paper is quite wide, considering using a narrower one. I think some details are not so clear in this setting, though I can tell your result is better than others.

**Limitations:**

1. Although the author used some way to speed up the process both for training and sampling. However, diffusion models are still slow compared to other methods both traditional and deep learning and non-diffusion way.

---

> ### Author Rebuttal · Authors · 2024-08-07
>
> We thank the reviewer for providing the encouraging feedback! Below, we address the concern:
>
> > **Q:** *Discuss related work with other regularization methods*
>
> A:
> We are aware of other works that apply external regularization between 2D slices with diffusion models such as [1,2] as we cited in our paper. Some other works train a xy plane prior and a yz plane prior [3] (also cited) for medical image reconstruction. Nevertheless, to the best of our knowledge, our work is the first one that uses only one 3D generative prior for medical image reconstruction without any additional regularization or priors. We will add more discussions on related works in our camera-ready version.
>
> > **Q:** *Release code*
>
> A:
> Thank you for the encouraging feedback! We are in the process of refactoring our code, and will release it ASAP after acceptance.
>
> > **Q:** *Difference between the score-med paper*
>
> A: The problems we solve and our motivations are different from the score-med paper [4]. The score-med paper is one of the first works that leverages a 2D diffusion prior for solving 2D medical image reconstruction problems. Our work is the first work that investigates whether using a 3D patch diffusion prior can lead to improvement over a 2D diffusion prior for 3D medical image reconstruction. The answer to the question is positive as demonstrated in our paper. Also our baseline "DDS 2D" is a very similar approach to the score-med paper. We show that our method outperforms DDS 2D by a significant margin.
>
>
> > **Q:** *Window center and window width*
>
> A: We follow [1,2,3,4]'s approach for processing CT images, to see bone structures better. Our window center is 100 HU, and window length is 2800 HU ([-1300 HU, 1500 HU]). In the rebuttal pdf, we also provide the visualization of our reconstruction in a much narrower window [-150 HU, 250 HU], and we observe that our method can reconstruct the fine details and structures of images very accurately with more projections.
>
>
> > **Q:** *However, diffusion models are still slow compared to other methods both traditional and deep learning and non-diffusion way.*
>
> A: We demonstrate that our method and other diffusion-based approaches can be faster than TV-based iterative traditional approaches (SBTV (one of the best non-deep-learning approach)) in our rebuttal pdf. The reason is the computation of 3DTV can be costly, but our approach does not need any external regularization so it can be faster.
>
>
>
> [1] Chung, Hyungjin, et al. "Solving 3d inverse problems using pre-trained 2d diffusion models." Proceedings of the IEEE/CVF Conference on Computer Vision and Pattern Recognition. 2023
>
> [2] Chung, Hyungjin, Suhyeon Lee, and Jong Chul Ye. "Decomposed Diffusion Sampler for Accelerating Large-Scale Inverse Problems." The Twelfth International Conference on Learning Representations.
>
> [3] Lee, Suhyeon, et al. "Improving 3D imaging with pre-trained perpendicular 2D diffusion models." Proceedings of the IEEE/CVF International Conference on Computer Vision. 2023.
>
> [4] Song, Yang, et al. "Solving Inverse Problems in Medical Imaging with Score-Based Generative Models." International Conference on Learning Representations.

---

> ### Author Response · Authors · 2024-08-14
> **Looking for your feedback and input**
>
> Dear Reviewer FN1q,
>
> We really appreciate your time for writing the review and providing the encouraging feedback. We want to highlight some novelties in our algorithm design in additional to our motivations and problem settings as the author-reviewer discussion window is closing.
>
> The key novelty in our paper mainly focuses on designing an efficient 3D generative prior by learning from the 3D volume for CT reconstruction. We will revise our paper to highlight the our key novelty, and clearly state our contribution. In our paper, we have the several key novel designs:
>
> 1. stack each 2D slice and each adjacency slice of a 3D medical image as a multi-channel image, and use a diffusion prior to learn the distribution of the patch of the adjacency slices. To our best knowledge, we are the **first** to propose this learning strategy in medical image reconstruction.
> 2. Propose a random blending algorithm for randomly partitioning and blending adjacency patches, which demonstrates to significantly improve slice inter-smoothness. To our best knowledge, we are the **first** to propose this method for medical image reconstruction.
> 3. ***More importantly***, we propose a novel jumping-slice patch idea, which enable our diffusion prior to learn the long-range dependency instead of focusing on adjacency slices. We treat non-adjacency slices as a wide patch and learn the distribution of 3D patches with different thickness. This method enables us to work on larger volumes. To our best knowledge, we are the **first** to propose this method. We demonstrate that we can directly learn the long-range dependency via jumping slices. Results show this design significantly reduces image artifacts as demonstrated in Figure.4 in our main paper.
>
> The novelty of the score-med paper [1] lies in its inverse problem solving technique and posterior sampling, which is not the main focus of our paper. We are more focusing on learning the 3D prior distribution, and we propose several novel designs to achieve this goal. Thanks for your consideration again, hope this comment resolves some of your concerns. Feel free to let us know if there is any remaining question about the manuscript and we will try our best to answer.
>
> [1] Song, Yang, et al. "Solving Inverse Problems in Medical Imaging with Score-Based Generative Models." International Conference on Learning Representations.

---

> > ### Comment · Reviewer_FN1q · 2024-08-14
> >
> > I think the author's reply solved my doubts, I will improve my score. I suggest that the git repo be added to the final version, regardless of whether the paper is accepted.

---

### Official Review · Reviewer_Vnw7 · 2024-07-09

**Soundness:** 1
**Presentation:** 3
**Contribution:** 3
**Rating:** 6
**Confidence:** 4

**Summary:**

1. The method proposed a method that learns the 3D-patch image prior incorporating the cross-slice dependency.
2. The method achieves state-of-the-art reconstruction results for 3D volumetric imaging for the task of ultra-sparse-view and limited-angle
3D CT reconstruction on  "AAPM 2016 CT challenge" dataset and "LIDC-IDRI" dataset.

**Strengths:**

1. The probabilistic modeling of the 3D volume taking into account neighboring slices is novel.
2. The paper describes the author's motivation and implementation plan very well.

**Weaknesses:**

1. Sparse angle CT imaging is typically defined as using fewer than 100 angles. The paper should evaluate the method's performance at various angles (e.g., 20, 40, 60, 80, and 100 angles). Demonstrating that this method outperforms the comparison method in PSNR/SSIM at fewer than 10 angles does not necessarily indicate better performance at other sparse angles (e.g., 20, 40, 60, 80, and 100 angles).

2. The comparison should include classic CT reconstruction methods, not just FBP, as FBP lacks prior information and performs poorly among traditional methods. Suggested comparison methods include:
2.1 SIRT (Simultaneous Iterative Reconstruction Technique) algorithm
2.2 Conjugate Gradient Least Squares (CGLS) algorithm
2.3 Split-Bregman (SB) Total Variation: Goldstein, T. and Osher, S., 2009. The split Bregman method for L1-regularized problems. SIAM journal on imaging sciences, 2(2)
Because the traditional iterative method also involves the adjustment of hyperparameters, for a fair comparison, please first use grid search to search the hyperparameters of the traditional method to obtain the best performance of the traditional method, and then compare the performance of the traditional method with the performance of this paper. Also, please report the extent of the grid search. Only in this way can it can be fully evaluated in the numerous algorithms for CT image reconstruction.

3. Figure 3 shows a large number of structures that do not exist in CT images, which will interfere with the doctor's diagnosis. Please explain the reasons for this phenomenon and how to avoid it. (For example, increase the number of angles, which is why I am very concerned about increasing the angles). If increasing the angles can avoid the artifacts that come with the generative model on a large scale, the potential for medical applications will increase.

4. As the abstract said, the method's purpose is to decrease the cost of memory, time. In order to prove that this method has achieved the original intention of it, please show the time consuming and memory consuming during training phase and testing phase. And compare the memory and time consuming between different methods.

**Questions:**

1. If the "conditioned slices" in equation (2) increase("conditioned slices" is: x[:,:,i-j:i-1] and x[:,:,i+1:i+j]), will the numerical performance better? If it is better, how does the performance gain change as the number of adjacent slices increases?

2. The paper mentions using a different partition so that the previous border slices can be included in another partition. However, if the method creates a new partition and then uses this new partition to create 3D patches, the adjacent slices of different 3D patches still cannot be updated simultaneously when the algorithm attempts to update the slices based on the scores calculated in each 3D patch. So, my question is, if the method uses the joint distribution modelling method (Eq. (5)), how can it avoid the situation I mentioned?

3.  For the experimental part,  The reconstruction mentioned here is the reconstruction from the simulated projection to the 3D CT volume or the reconstruction  from  the real CT projection(real measurement) to the 3D CT volume? If it is a simulated projection, is simulated noise added to the simulated projection? (Note: The noise I am referring to here is not the noise added in the diffusion model to train the neural network, but the noise added to simulate the real projection, which is generally a combination of Poisson noise and Gaussian noise.)

**Limitations:**

The artifacts generated by the diffusion model may lead to misjudgment in the doctor's diagnosis. Please provide methods to avoid artifact, such as increasing the angle or the amount of training data?

---

> ### Author Rebuttal · Authors · 2024-08-06
>
> Thank you for providing the valuable feedback. Below, we address the concern:
> First, we address concerns regarding to additional experiments:
>
> > **Q:** *Evaluate method's performance at various angles.*
>
> A:
> - We provide results on [20, 40, 60, 80, 100] views for our method (DiffusionBlend++) as well as baselines with those angles (including the SB(Split-Bregman)TV, CGLS, and SIRT baselines) for the LDCT dataset in the rebuttal pdf as well in the table below. Results show that our method ***outperforms the baselines significantly*** for ***every*** angle we evaluated on. From Figure 1,2,3 in the rebuttal pdf DiffusionBlend++ starts to reconstruct images very close to the ground truth with 20 projections or more, but other baselines such as SIRT and CGLS still struggle to get a satisfying reconstruction with 60 views or more. The following table shows the reconstruction performance on the coronal plane of different methods. We observe that DiffusionBlend++ (ours) has a significant margin above baselines for every view. Note that DiffusionBlend++ still outperforms DDS2D significantly with >40 views, which demonstrates that our 3D prior is still very useful even with much more views.
> - The motivation of this paper is to exploit the 3D data-driven prior (instead of external 3D regularization or learning another 2D prior on another plane) for medical image reconstruction. The hypothesis is that with the aid of the 3D prior, we can have better reconstruction quality comparing to only using 2D priors. Many previous works leverage 2D diffusion priors for sparse-view reconstructions ( > 10 views), but only a few works exploit reconstructions with fewer than 10 views. By this consideration, we do not provide results for more than 10 projection in the main paper. Nevertheless, we will add additional results with more projection angles and more traditional baselines in the camera-ready version.
> ``Table.1 Coronal plane results on LDCT test set``
> | Method           |4 views| 6 views| 8 views | 20 views | 40 views | 60 views | 80 views | 100 views |
> |------------------|----------|----------|----------|----------|----------|----------|-----------|-----------|
> | FBP |    11.16/0.236   |    13.18/0.268    |    14.64/0.325   |    20.22/0.594    |     26.09/0.792    |     29.97/0.886    |     32.89/0.933    |     35.16/0.959    |
> | SIRT |    22.12/0.823    |    23.54/0.841    |    24.50/0.851    |    27.72/0.882    |     30.82/0.918    |     33.34/0.945    |     35.56/0.962    |     37.66/0.974    |
> | CGLS |    22.11/0.823    |    23.49/0.841    |    24.44/0.850   |    27.67/0.882    |     30.66/0.918    |     32.96/0.944    |     34.82/0.960    |     36.36/0.971    |
> | SBTV |    23.33/0.849    |    25.83/0.881    |    27.85/0.896    |    31.51/0.938   |     34.83/0.966    |     36.74/0.963    |     37.59/0.973    |     38.97/0.979    |
> | DDS2D |    30.25/0.916    |    32.33/0.939    |    33.64/0.950   |    35.34/0.961    |     38.26/0.977    |     40.07/0.984    |     41.19/0.987   |     41.88/0.989    |
> | DDS |    30.89/0.932   |    32.95/0.879    |    33.97/0.935    |    35.81/0.967    |     37.59/0.975    |     38.26/0.978    |     39.07/0.980    |     39.54/0.983    |
> | DiffusionBlend++ |    ***34.27/0.955***    |    ***36.66/0.963***   |    ***37.87/0.968***    |   ***40.56/0.980*** |  ***42.53/0.987***    |     ***43.66/0.990***    |     ***44.42/0.992***    |     ***44.95/0.993***    |
>
>
>
>
>
> > **Q:** *Comparison with classic CT reconstruction methods, not just FBP*
>
> A: Table.1 and Figure.1,2,3 in the rebuttal pdf show more results on additional classic baselines. Note that DiffusionBlend++ ***does not need to tune any hyperparameter*** when the number of projections changes, but this is a requirement for some traditional methods.
> - SBTV: We implement this algorithm with variables splitting of 3D anisotropic TV regularization (Dz, Dx, and Dy). We first check number of iterations, note that the performance converges with around 30 iterations. We did a grid search of hyperparameters on 9 validation images (not in the test set) for every projection angles. For example: the searching for 20 projections is conducted as below:
>
> | $\lambda$       |  $\lambda / \rho = 0.02$ | $0.04$ | $0.08$ | $0.16$ |
> |--------|------|------|------|------|
> | 1.25   |   27.88   | 28.34   |  28.30    |  27.88    |
> | 2.5    |  28.16    |  ***28.36***    |  28.07    |  27.44    |
> | 5      |  28.10    | 28.06     |  27.57    |   26.73   |
> | 10     |   27.98   |  27.57    | 26.77     |  26.24    |
>
> - SIRT: This algorithm iteratively updates the reconstruction based on the residual between projection of the reconstruction and the GT. It only has the number of iterations as its hyper-parameter. We note that during inference, PSNR increases with more iterations, but saturates later. So we set the total number iterations to be 1000, with an early stopping threshold of 1e-6 between two consecutive iterations.
>
> - CGLS: This algorithm uses conjugate gradient for solving least square problems. In our case, we use $CG(A^TA + \rho x^Tx, A^Ty)$, $\rho$ is set to be 1e-4 based on grid search for numerical stability. We tune the number of iterations on validation set, and find that performance saturates at around 25 iterations.
>
> > **Q:** *Whether the projections have noise?*
>
> A: Following approaches in [1,2,3,4,5], we do not add noise in our projections since the noise is very small for normal dose. Nevertheless, we also run additional experiments with 8-view reconstruction for LDCT test set in the low-dose setting with Poisson-Gaussian noise. The pre-log noise model is given by $y_i = \text{Poisson}(I_0 \exp(-Ax_i)) + N(0, \sigma^2)$. Following [6] to account for same dose level, we set $I_0 = 10^6$, $\sigma = 5$. Results as below show that our method is robust to noise.
> |        |   coronal|  sagital | axial|
> |-----|----|-----|-----|
> | no noise | 37.87/0.968| 36.48/0.968| 35.69/0.966|
> |with noise| 37.62/0.966| 36.27/0.965 | 35.47/0.964|

---

> > ### Comment · Reviewer_Vnw7 · 2024-08-10
> >
> > I think the author's reply solved some of my doubts, I will improve my score. I suggest that the supplementary experiments be added to the final version, regardless of whether the paper is accepted or not.

---

> > > ### Author Response · Authors · 2024-08-10
> > >
> > > Thanks for the review and reading our rebuttal. Your review is crucial for us to improve our manuscript. We will definitely add all the supplementary experiments discussed in the rebuttal period to our paper. Feel free to let us know if there is any remaining question about the manuscript and we will try our best to answer.

---

> ### Author Response · Authors · 2024-08-07
> **Rebuttal (Part 2)**
>
> > **Q:** *Explain why wrong structures exist in the reconstruction*
>
> A:
> - With ultra-sparse views (such as 4 views), CT reconstruction is extremely difficult since very few measurements are taken, so many traditional reconstruction algorithms completely fail in this scenario as demonstrated in Figure 3. Also, non-deep learning methods generate lots of structures that do not exist even with as much as 20 views as demonstrated in Figure 2 in the rebuttal pdf. To account for this missing information with ultra-sparse views, DiffusionBlend(++) generates many image structures that represent the ground truth image well, and some other structures that may look different from ground truth, due to the extremely big challenge of this ultra-sparse view reconstruction task. Note that our method is still significantly better than all compared baselines in such a challenging task.
> - With increasing number of views, more measurements are taken, and it is more likely we can reconstruct accurate structures (as mentioned by reviewer Vnw7). We provide reconstruction examples in the rebuttal pdf; results show that the image structures look ***almost identical*** to the ground truth with 40 views or more.
>
>
> > **Q:** *The method's purpose is to decrease the cost of memory, time. Show the time consuming and memory consuming*
>
> A:
> -  Firstly, the main purpose of this work is, for the first time, to investigate how to learn a 3D patch diffusion prior that can improve ultra-sparse 3D CT reconstruction performance. The abstract mentions the difficulty of training a diffusion model on 3D full volumetric data (e.g. simply cannot fit into a 48G A40 GPU), which motivates us to train a 3D patch diffusion model which is much more computational efficient than a full 3D model, while achieving impressive results.
> -  We have provided the inference time of our method and other baselines in Table 9 of our paper on the 500 slices of the LDCT test set. Additionally, we also provide the training time, training memory, inference memory in the rebuttal pdf. Overall our method is efficient since it only uses a 3-channel diffusion model. ***Surprisingly***, our method has better inference time than SBTV, while do not require much more memory than SBTV.  The reason is that unlike SBTV which requires expensive 3D TV computation, our method does not require any external regularizations, so the inference speed is faster.
>
> | Method | Training memory | Training time | Inference memory | Inference time |
> |------| ------| ------| ------| ------|
> |SBTV [7] | - | -| 6045MB | 62 mins |
> |DiffusionMBIR [1] | 29043MB | 47 hours | 22062MB | 23 hours |
> |DDS [2] |  27040MB | 12 hours | 20608MB | 48 minutes |
> |DiffuisonBlend++ (Ours) | 35384MB | 4.5 hours | 9976MB | 32 minutes |

---

> ### Author Response · Authors · 2024-08-07
> **Rebuttal (Part 3)**
>
> > **Q:** *Whether more conditional slides will make numerical performance better?*
>
> A: We perform experiments that train our conditional model (DiffusionBlend) on 0, 2, 4, 6 conditional slices and evaluate on the LIDC dataset with 8 projections. Results show that including conditional slices increases the performance significantly, but adding more conditional slices is not guaranteed to further improve the performance and the improvement may be marginal.
> To explain why the improvement is marginal with more conditioning slices, we can think about the iterative process of diffusion reverse sampling [1,2]. For example, if using 2 conditional slices, for one specific slice, the first iteration conditions on 2 neighboring slices, but its neighbors also conditions on 2 additional slices. At the second iteration, that slice still conditions on its neighbors (which already had information from 2 additional slices), so it can get the information from the condition of 4 slices, by tracing back to the first iteration. In this way, we can show that with sufficient number of iterations, one specific slice can take the information (condition) on every other slice. Since we have sufficient number of iterations, either 2,4 or 6 conditional slices enables us to condition one slice on all other slices eventually.
>
> This method enables us to learn the 3D prior very ***efficiently*** with minimal number of conditioning slices.
> | Conditioning slices        | PSNR/SSIM (8 views)|
> |------| ------|
> |0 | 30.98/0.894|
> |2 | 33.73/0.933|
> |4 | 33.32/0.932 |
> |6 | 33.53/0.936 |
>
> > **Q:** *The adjacent slices of different 3D patches still cannot be updated simultaneously*
>
> A: Previous work [1,3] demonstrate that each step of summation, gradient descent, or even ADMM updates can be split into each step of diffusion reverse sampling process while achieving satisfying performance. Here even though the adjacency slices may not be updated simultaneously at consecutive reverse sampling iterations, it is similar to a Monte Carlo average of the score of the distribution of different partitions as demonstrated in Eq.7. So the goal is not to compute the score of the 3D patch distribution as in Eq.5, but to approximate the score of the ground truth distribution p(x) by averaging the scores of the distributions of multiple partitions.
>
> [1] Chung, Hyungjin, et al. "Solving 3d inverse problems using pre-trained 2d diffusion models." Proceedings of the IEEE/CVF Conference on Computer Vision and Pattern Recognition. 2023
>
> [2] Chung, Hyungjin, Suhyeon Lee, and Jong Chul Ye. "Decomposed Diffusion Sampler for Accelerating Large-Scale Inverse Problems." The Twelfth International Conference on Learning Representations.
>
> [3] Lee, Suhyeon, et al. "Improving 3D imaging with pre-trained perpendicular 2D diffusion models." Proceedings of the IEEE/CVF International Conference on Computer Vision. 2023.
>
> [4] Song, Yang, et al. "Solving Inverse Problems in Medical Imaging with Score-Based Generative Models." International Conference on Learning Representations.
>
> [5] Chung, Hyungjin, et al. "Improving diffusion models for inverse problems using manifold constraints." Advances in Neural Information Processing Systems 35 (2022): 25683-25696.
>
> [6] Ye, Siqi, et al. "SPULTRA: Low-dose CT image reconstruction with joint statistical and learned image models." IEEE transactions on medical imaging 39.3 (2019): 729-741.
>
> [7]  Goldstein, Tom, and Stanley Osher. "The split Bregman method for L1-regularized problems." SIAM journal on imaging sciences 2.2 (2009): 323-343.

---

### Official Review · Reviewer_YiqS · 2024-07-11

**Soundness:** 3
**Presentation:** 3
**Contribution:** 3
**Rating:** 5
**Confidence:** 4

**Summary:**

This paper proposes a novel method for learning 3D diffusion priors for CT reconstruction, which does not require large-scale data or computational resources. It presents two approaches: DiffusionBlend and DiffusionBlend++. The former learns a specific frame given adjacent slices, while the latter learns a 3D patch. Experimental results demonstrate that both methods are efficient and outperform previous works.

**Strengths:**

1. This method enables 3D reconstruction without the need for large-scale data and resources.
2. It achieves excellent inter-slice consistency without relying on external regularizations.
3. Additionally, this method outperforms existing baselines.

**Weaknesses:**

The only concern is that this method may be too simplistic to be extended to a broader field. For example, I believe that inter-slice smoothness cannot be guaranteed if this method is applied to videos. However, given that this method is proposed for CT reconstruction, I don't think this issue warrants the rejection of the paper.

**Questions:**

I don't have questions about this paper.

**Limitations:**

The authors have discussed the limitations.

---

> ### Author Rebuttal · Authors · 2024-08-06
>
> Thank you for providing the valuable feedback. Below, we address the concern:
>
> > **Q:** *Extending the method to a broader field*
>
> A: Our method is flexible and does not assume any specific data modality and forward models, so it should work for other modalities such as natural 3D images or videos, and for other inverse problems as well. Nevertheless, the results on other modalities and tasks are beyond the scope of this work, but we tested unconditional generation of 3D CT images which show good generation quality and inter-slice smoothness. We will include this experiment in the camera-ready version as reference. Thus, we think that the inter-slice smoothness of the learned prior from our method could be potential to work for videos as well to obtain a inter-frame coherence, which can be a very interesting future works. We will add this discussion in the camera-ready version as well.

---

> > ### Comment · Reviewer_YiqS · 2024-08-10
> >
> > Thank you for your response; it has addressed some of my concerns to a certain extent. However, I will still maintain my score

---

> > > ### Author Response · Authors · 2024-08-13
> > > **Further clarifications on more general applications of our method**
> > >
> > > Dear Reviewer YiqS,
> > >
> > > We present some additional preliminary analysis on whether our method is still feasible on video data, and the results look promising. Hope that this new evidence can be helpful to answer your question.
> > >
> > > To further address your concern about a broader application of the proposed method, we first compute the inter-slice consistency for unconditional generation of 3D CT generation with / without random blending. In addition, we also perform more experiments on unconditional video generation to see whether our algorithm can maintain inter-slice (frame) consistency for video data.
> > >
> > > We tested on the Sky Time-lapse dataset [1], which has time sequences of sky images. We use the diffusion model pretrained with 256x256 ImageNet data, and then fine tuned on 20 video sequences. Since our method can only handle single channel images like medical images (but should be able to extend to RGB with a little effort), we only use the first channel of the video for fine tuning). The learning rate is set to be the same as CT images, and we observe convergence at 5000 iterations with a batch size of 4. Then we test the inter-slice total-variation (TV) value on the unconditional generation of a 16-frame video (with / without random blending).
> > >
> > > One key observation is that video data has more coherent frames so it is actually easier for us to train the 3D patch diffusion on it since there is no inter-slice jump which is observed on 3DCT data.  We present the z-axis TV values in the table below, which is computed by $||D_z(x)||_1/n$, where $n$ is number of pixels. The ground truth is taken from the test dataset for CT, and an average of 10 videos in sky-timelapse video. The results are based on an average from 10 generated volumes/videos.
> > > | Task | Without Blending | With Blending | Ground Truth |
> > > |------| ------| ------| ------|
> > > |3DCT Generation | 0.0236 | ***0.0035*** | 0.0065
> > > |Skylapse Video Generation | 0.0419 | ***0.0021*** | 0.0036  |
> > >
> > > We observe with the proposed blending method improves the inter-slice smoothness significantly for both video and 3DCT data. Some video data has smaller inter-slice variation than 3D medical images. We observe that our method can generate 3D images/videos with the inter-slice smoothness comparable to the ground truth. Thus, this prior can also be useful for solving inverse problems in a broad applications.
> > >
> > >
> > > Thanks for the review and reading our rebuttal. Your review is crucial for us to improve our manuscript. Feel free to let us know if there is any remaining question about the manuscript and we will try our best to answer.
> > >
> > > [1] Zhang, Jiangning, et al. "Dtvnet: Dynamic time-lapse video generation via single still image." Computer Vision–ECCV 2020: 16th European Conference, Glasgow, UK, August 23–28, 2020, Proceedings, Part V 16. Springer International Publishing, 2020.

---

### Author Rebuttal · Authors · 2024-08-06

Firstly, we would like to sincerely thank the reviewers for taking the time to review our paper and providing constructive feedback. We are encouraged that the reviewers think that in our paper
- ```The first diffusion model in CT considered z-axis consistency and used 3d data as neural network input. A SOTA results with astonishing visualization and metrics results.``` (Reviewer  FN1q),
- ```This paper proposes a novel method. Experimental results demonstrate that both methods are efficient and outperform previous works.``` (Reviewer YiqS),
- ```The probabilistic modeling of the 3D volume taking into account neighboring slices is novel. The paper describes the author's motivation and implementation plan very well.``` (Reviewer Vnw7).

To address the concerns shared by some reviewers, we would like to globally highlight several key points in our response.

***1. This method outperforms the comparison method in PSNR/SSIM at fewer than 10 angles does not necessarily indicate better performance at other sparse angles, more traditional baseline results***

- We provide results on [20, 40, 60, 80, 100] views for our method (DiffusionBlend++) as well as baselines with those angles (including the SB(Split-Bregman)TV, CGLS, and SIRT baselines requested by Reviewer Vnw7) for the LDCT dataset in the rebuttal pdf and in the table below. Results show that our method ***outperforms the baselines significantly*** for ***each different number of angles*** we evaluated on. From Figure 1,2,3 in the rebuttal pdf, DiffusionBlend++ is able to reconstruct images very close to the ground truth with 20 projections or more, but other baselines such as SIRT and CGLS still struggle to get a satisfying reconstruction with 60 views or more.
- The motivation of this paper is to exploit the 3D data-driven prior (instead of external cross-slice regularization or learning 2D priors on different planes) for medical image reconstruction. The hypothesis is that with the aid of the 3D prior, we can have better reconstruction quality compared to only using 2D priors. Many previous works leverage 2D diffusion priors for sparse-view reconstructions ( > 10 views), but only a few works exploit reconstructions with fewer than 10 views. By this consideration, we do not provide results for more than 10 projection in the main paper. Nevertheless, we will add additional results with more projection angles and more traditional baselines in the camera-ready version.


***2. A large number of structures that do not exist in CT images, which will interfere with the doctor's diagnosis***


- With ultra-sparse views (such as 4 views), CT reconstruction is extremely difficult since very few measurements are taken, so many traditional reconstruction algorithms completely fail in this scenario as demonstrated in Figure 3. Also, non-deep learning methods generate lots of structures that do not exist even with as high as 20 views as demonstrated in Figure 2 in the rebuttal pdf. To account for this missing information with ultra-sparse views, DiffusionBlend(++) generates many image structures that represent the ground truth image well, and some other structures that may look different from ground truth. Nevertheless, it is still significantly better than all baselines considered in this paper.
- With increasing number of views, more measurements are taken, and it is more likely we can reconstruct accurate structures (as mentioned by reviewer Vnw7). We provide reconstruction examples in the rebuttal pdf; results show that the image structures look ***almost identical*** to the ground truth with 40 views or more.

***3. Difference from other related works***
- The problems we solve and our motivations are different from the score-med paper [1]. The score-med paper is one of the first works that leverages a 2D diffusion prior for solving ***2D*** medical image reconstruction problems. Our work is ***the first work*** that investigates whether using a 3D patch diffusion prior can lead to improvement over a 2D diffusion prior for ***3D*** medical image reconstruction. The answer  to the question is ***positive*** as demonstrated in our paper. Also our baseline "DDS 2D" is a very similar approach to the score-med paper. We show that our method outperforms DDS 2D by a significant margin.
- Some other works use inter-slice external regularization, such as total variation (TV) [2] or multiple 2D priors on different planes for reconstruction [3], which we have both cited in our paper, as mentioned by reviewer FN1q. Other methods such as ADMM-TV uses the TV regularization for xy, xz, and yz plane. Nevertheless, our work differs from them fundamentally in that our work ***for the first time*** learns one generative 3D prior and uses only that prior without any other regularization for medical image reconstruction. We will include more discussions about related works in our camera-ready version.


***4. Discussion on Computational Efficiency***

We have provided the inference time of our method and baselines in Table 9 of our main paper.  In addition, we also provide training time, training memory, inference memory and inference time in the rebuttal pdf. Results show that we achieve better inference memory efficiency and decent inference speed compared to DDS since we do not use the computational-heavy total variation regularization (while DDS claims to be one of the most efficient diffusion solvers for medical imaging). ***Surprisingly***, we find the non-deep-learning method SBTV has a worse inference efficiency compared to our method due to the heavy computational cost of 3D total variation optimization.

[1] Song, Yang, et al. "Solving Inverse Problems in Medical Imaging with Score-Based Generative Models." ICLR2022

[2] Chung, Hyungjin, et al. "Solving 3d inverse problems using pre-trained 2d diffusion models." CVPR2023

[3] Lee, Suhyeon, et al. "Improving 3D imaging with pre-trained perpendicular 2D diffusion models."ICCV2023

---

### Comment · Area_Chair_Jwyf · 2024-08-08
**Paper discussion and rating finalization**

Dear Reviewers,
Can you please have a look at the reports of the other reviewers and also the rebuttal from the authors and respond to their questions, if available, then discuss any issues of your concern, finalise and reach a consensus about the rating of the paper before the deadline of the next Tuesday, 13th August?
Thank you very much for your time, effort and contribution to the organization of NeurIPS 2024,
AC

---

### Comment · Area_Chair_Jwyf · 2024-08-11
**Reminder of the deadline**

Dear Reviewers,
While the deadline, Tuesday, 13th August, is approaching, can you please check at your early convenience the rebuttal from the authors, make or require further clarifications, if necessary, and interact with the authors over any further issues/concerns you may still have, and finalise the rating of the paper soon. Even though you have no further issues/concerns, you may want to acknowledge the responses to your questions in the rebuttal from the authors.
Thank you very much for your time, effort and contribution to the organization of NeurIPS 2024,
AC

---

### Decision · Program_Chairs · 2024-09-25

**Decision:**

Accept (poster)

**Comment:**

Overall, all the reviewers are interested in the topics, have good opinions towards the proposed method and the experimental results, and have positive rating towards and tend to accept the paper. Most concerns from the reviewers have been well addressed by the authors to largely the satisfactions of the reviewers with more discussions, clarifications and experimental results.